# BMPs direct sensory interneuron identity in the developing spinal cord using signal-specific not morphogenic activities

Madeline G Andrews[1,2,3], Lorenzo M del Castillo[1,3,4], Eliana Ochoa-Bolton[1,3,4], Ken Yamauchi[1,3], Jan Smogorzewski[5], Samantha J Butler[1,3]*

[1]Department of Neurobiology, University of California, Los Angeles, United States; [2]Neuroscience Graduate Program, University of California, Los Angeles, United States; [3]Eli and Edythe Broad Center of Regenerative Medicine and Stem Cell Research, University of California, Los Angeles, United States; [4]CIRM Bridges to Research Program, California State University, Northridge, United States; [5]Department of Dermatology, University of Southern California, California, United States

**Abstract** The Bone Morphogenetic Protein (BMP) family reiteratively signals to direct disparate cellular fates throughout embryogenesis. In the developing dorsal spinal cord, multiple BMPs are required to specify sensory interneurons (INs). Previous studies suggested that the BMPs act as concentration-dependent morphogens to direct IN identity, analogous to the manner in which sonic hedgehog patterns the ventral spinal cord. However, it remains unresolved how multiple BMPs would cooperate to establish a unified morphogen gradient. Our studies support an alternative model: BMPs have signal-specific activities directing particular IN fates. Using chicken and mouse models, we show that the identity, not concentration, of the BMP ligand directs distinct dorsal identities. Individual BMPs promote progenitor patterning or neuronal differentiation by their activation of different type I BMP receptors and distinct modulations of the cell cycle. Together, this study shows that a 'mix and match' code of BMP signaling results in distinct classes of sensory INs.
DOI: https://doi.org/10.7554/eLife.30647.001

*For correspondence:
butlersj@ucla.edu

Competing interests: The authors declare that no competing interests exist.

## Introduction

A common theme in organ development is the existence of signaling centers: restricted sources of inductive growth factors that pattern the identity of surrounding tissues. A notable example of this mechanism is found in the developing spinal cord, which contains two signaling centers, the floor plate (FP) at the ventral midline and the roof plate (RP) at the dorsal midline (*Butler and Bronner, 2015*). The FP secretes sonic hedgehog (Shh), which has been proposed to act as a morphogen to specify neuronal identity (*Briscoe and Ericson, 2001*). In this model, neural progenitors adopt their specific identities in response to the local concentration of Shh, which depends on their position with respect to the FP. Progenitors immediately adjacent to the FP become the ventral-most interneurons (INs), while cells progressively further away are specified as motor neurons, and multiple classes of ventral INs (*Briscoe and Ericson, 2001*). More recent studies have suggested that Shh also acts as a temporal morphogen, such that progenitor identity depends on the duration of exposure to Shh (*Kong et al., 2015*; *Dessaud et al., 2010*). Together, these studies have been highly influential, both demonstrating that Shh functions as a textbook example of a morphogen and providing a general mechanism by which inductive signaling centers may specify an array of cellular fates.

The RP secretes two families of growth factors, the Bone Morphogenetic Proteins (BMPs) (*Liem et al., 1995*) and Wnts (*Parr et al., 1993*). Wnts are thought to act as mitogens, generally promoting dorsal progenitor proliferation (*Megason and McMahon, 2002*), while the BMPs have been proposed to act as morphogens, patterning dorsal progenitors in a concentration dependent manner similar to Shh (*Lee and Jessell, 1999*). Studies examining the loss (*Hazen et al., 2012*; *Lee et al., 1998*; *Wine-Lee et al., 2004*; *Le Dréau et al., 2012*; *Nguyen et al., 2000*) or gain (*Yamauchi et al., 2008*; *Timmer et al., 2005, 2002*; *Chizhikov and Millen, 2004*) of BMP signaling in vivo have shown that BMP signaling is critical for the formation of the dorsal interneuron (dI) 1, dI2 and dI3 classes of sensory neurons. However, there has been only limited evidence that the BMPs act as morphogens in vertebrates. Experiments analogous to those performed for Shh, demonstrated that conditioned medium from BMP4-tranfected COS cells was sufficient to specify dI1s in chicken neural plate tissue explants (*Liem et al., 1997*). However, treatment of these explants with diluted BMP4-conditioned medium was not particularly effective at inducing dI3s, the predicted result if BMP4 functions as a morphogen. A more recent study (*Tozer et al., 2013*), suggested there is a temporal component to BMP4 signaling, such that the longer exposure to BMP4 results in more dorsal progenitor identities in chicken neural plate explants. However, it was not determined how this manipulation affected post-mitotic neural identity.

Moreover, none of these studies addressed a key difference between signaling in the RP and FP: there are multiple members of the BMP family present in the dorsal spinal cord, including BMP4, BMP5, BMP6, BMP7 and Growth/Differentiation Factor (GDF) 7 (BMP12). It has remained unresolved how these different factors might cooperate to establish a unified morphogen gradient. An alternative model is that different BMPs have signal-specific effects, such that individual BMPs are responsible for the induction of particular neural fates (*Lee and Jessell, 1999*). Supporting this hypothesis, loss of function mutations in *Gdf7* result in the specific ablation of the Lhx2[+] dI1A subpopulation in mouse (*Lee et al., 1998*), leaving the other dI populations intact. Similarly, knocking down *Bmp4* expression in the chicken reduces the number of dI1s, while the loss of *Bmp7* was unexpectedly shown to reduce the number of dI1s, dI3s and dI5s (*Le Dréau et al., 2012*). These findings support the hypothesis that different BMPs have non-redundant functions specifying dorsal cell fates, however they also contradicted previous analyses of *Bmp7*[-/-] mice that found no alterations in the number of dI1s (*Lee et al., 1998*; *Butler and Dodd, 2003*). Additionally, none of these studies evaluated the role of BMP concentration directing dorsal cell fate in vivo or resolved mechanistically how different BMPs might direct dorsal progenitors towards specific fates.

In this study, we have used *in ovo* electroporation of chicken spinal cords and mouse embryonic stem cell (mESC) cultures to methodically determine how the complement of dorsally expressed BMPs specifies neuronal identity. Both our in vivo and in vitro approaches support the model that the identity of the BMP ligand, rather its concentration, can direct a unique, and species-specific, range of dorsal cellular identities. We find that specific BMPs can promote either progenitor patterning or neuronal differentiation, possibly by their distinct modulations of the cell cycle. Furthermore, the ability to promote patterning or differentiation is mediated through activation of different type I BMP receptors (Bmprs). Together, this study provides insight into the mechanism by which a 'mix and match' code of BMP signaling results in the formation of the RP itself, and three distinct classes of sensory INs.

## Results

### Timeline of BMP expression in chicken embryos during neurogenesis

Multiple BMPs are present in the dorsal spinal cord (*Lee et al., 1998*; *Liem et al., 1997*), and BMP signaling has been shown to be critical for dorsal spinal identity (*Hazen et al., 2012*; *Wine-Lee et al., 2004*). However, the mechanism(s) by which different BMPs act to direct distinct dorsal IN identities remain unresolved. To address this question, we assessed the timing by which different BMPs are expressed in the chicken spinal cord (*Liem et al., 1997*), with respect to markers of dorsal patterning. Pax3, one of earliest general markers of dorsal spinal identity (*Mansouri and Gruss, 1998*), is expressed in all dorsal progenitors in the ventricular zone (VZ), prior to Hamburger-Hamilton (HH) (*Hamburger and Hamilton, 1992*) stage 14 (*Figure 1A*). Dorsal INs arise 12–24 hr after the onset of Pax3 expression. Dorsal interneuron (dI) 1 s are generated from the *Atoh1*[+] + progenitor

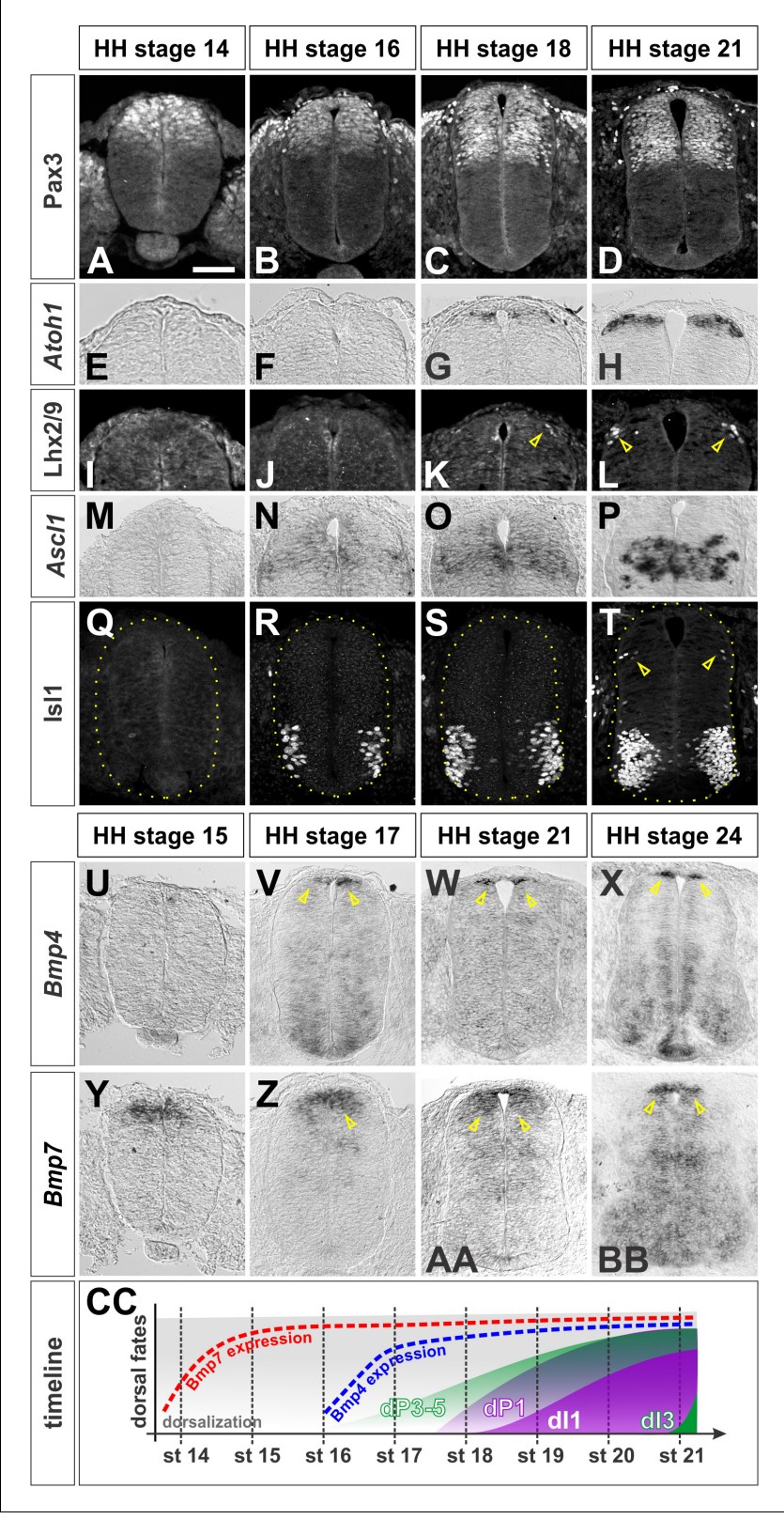

**Figure 1.** Timeline of dorsal patterning in the chicken spinal cord. Brachial (**A, B, E, F, G, H, I, J, K, L, M, N, Q, U, X, Y, BB**) or thoracic (**C, D, O, P, R, S, T, V, W, Z, AA**) level transverse sections from Hamburger-Hamilton (HH) stage 14–24 chicken spinal cords processed for immunohistochemistry (**A–D, I—L, Q–T**) or in situ hybridization (**E–H, M–P, U–BB**). (**A–D**) Pax3 is present in dorsal progenitors prior to HH stage 14 and persists throughout dorsal

*Figure 1 continued on next page*

*Figure 1 continued*

spinal cord patterning and differentiation. (**E–H**) *Atoh1* expression in dP1 progenitors begins prior to HH stage 18. (**I–L**) Lhx2/9$^+$ dI1 neurons start to be born at the brachial-most levels of HH stage 18 embryos. (**M–P**) *Ascl1* expression in dP3-5 progenitors begins prior to HH stage 16. (**Q–T**) Isl1$^+$ dI3 neurons are born starting from HH stage 21. (**U–X**) *Bmp4* is expressed in cells flanking the RP prior to HH stage 17 and stays tightly localized to this region in the dorsal spinal cord. (**Y–BB**) *Bmp7* expression starts at ~HH stage 14, and it is expressed more broadly in the dorsal-most spinal cord, including the RP. (**CC**)A timeline summarizing the onset of *Bmp* expression and generation of dI1 and dI3 neurons. Scale bar: 50 µm.

DOI: https://doi.org/10.7554/eLife.30647.002

The following figure supplement is available for figure 1:

**Figure supplement 1.** Expression of *Bmps* in chicken and mouse embryos.

DOI: https://doi.org/10.7554/eLife.30647.003

(dP) one domain (*Helms and Johnson, 1998*). *Atoh1* is expressed by HH stage 18 (*Figure 1G*), and dI1s start to be born at the brachial levels at the same stage (arrow, *Figure 1K*). In contrast, *Ascl1* expression, which defines the dP3-5 domain (*Helms et al., 2005*), starts at HH stage 16 (*Figure 1N*), but is not robust until HH stage 21 (*Figure 1P*), when the first post-mitotic dI3s are born (arrows, *Figure 1T*).

Of the BMPs known to be expressed in the chicken spinal cord (*Liem et al., 1997*) only *Bmp4* and *Bmp7* are expressed in the dorsal spinal cord before the onset of dorsal neurogenesis. *Bmp7* expression starts immediately prior to HH stage 15 (*Figure 1Y*), and is maintained in both the RP and the dorsal-most progenitors (arrows, *Figure 1Z*-1BB), while *Bmp4* is expressed specifically in the RP by HH stage 17 (arrows, *Figure 1V*). *Bmp5*, *Bmp6* and *Gdf7* are not expressed in the dorsal spinal cord at early stages (*Figure 1—figure supplement 1A,B,E,F* and data not shown). Many of the *Bmps* are subsequently expressed in the ventral spinal cord by HH stage 24 (*Figure 1X and B'*, S1D, S1H). Together, this timeline (*Figure 1CC*) suggests that BMP4 and BMP7 are the relevant BMPs that direct the specification of the dorsal INs in the chicken embryonic spinal cord.

## Bmp4 and Bmp7 direct different dorsal spinal fates in vivo

We next assessed how the BMPs direct dorsal IN identity by ubiquitously mis-expressing *Bmp4* and *Bmp7* throughout the chicken spinal cord by *in ovo* electroporation. We electroporated embryos at HH stage 15, immediately prior to dorsal progenitor patterning (*Figure 1CC*), and examined the consequence to dorsal cell fate two days later, at HH stage 25. During neurogenesis, the six dorsal progenitor domains, dP1 - dP6, differentiate into distinct populations of sensory INs, including dI1 - dI6 (*Figure 2A*) (*Butler and Bronner, 2015*). These populations can be distinguished by their different complements of transcription factors (*Figure 2B–C*).

Electroporation of either *Bmp4* or *Bmp7* results in high levels of phosphorylated (p), activated Smad1/5/8 (*Faure et al., 2002*) (*Figure 2E, F and V*), showing that both of these BMPs upregulate the BMP signaling pathway compared to control electroporations. However, BMP4 is far more effective, upregulating pSmad1/5/8 levels by ~65%, compared to ~30% for BMP7. Moreover, the ectopic expression of the two *Bmps* have different consequences for dorsal IN identity. Both BMP4 and BMP7 can increase the numbers of RP, dI1 and dI3 cells, however they do so to markedly different extents. *Bmp7* consistently increases the number of Mafb$^+$ RP cells (*Figure 2G–I and V*). In contrast, *Bmp4* has a variable effect on the number of Mafb$^+$ RP cells which does not rise to statistical significance, while it most effectively increases the number of Lhx2/9$^+$ dI1s (*Figure 2J–L and V*) and Isl1$^+$ dI3s (*Liem et al., 1997*) (*Figure 2P–R*). Only *Bmp4* is sufficient to increase the number of Lhx1/5$^+$, Pax2$^-$ dI2s (*Liem et al., 1997*; *Gross et al., 2002*) (*Figure 2M–O*). While the more-ventral dorsal IN populations have been shown to be independent of RP-signaling (*Lee et al., 2000*), we found that misexpression of *Bmp7* can modestly decrease the number of Pax2$^+$ dI4s (*Burrill et al., 1997*). These results suggest that BMPs direct dorsal patterning in signal-specific manner. Supporting this model, the misexpression of *Bmp5* and *Gdf7* also affects dorsal IN identity in a signal-specific way (*Figure 2—figure supplement 1*). Thus, each BMP has a unique range of effects on the specification of dorsal IN populations in vivo.

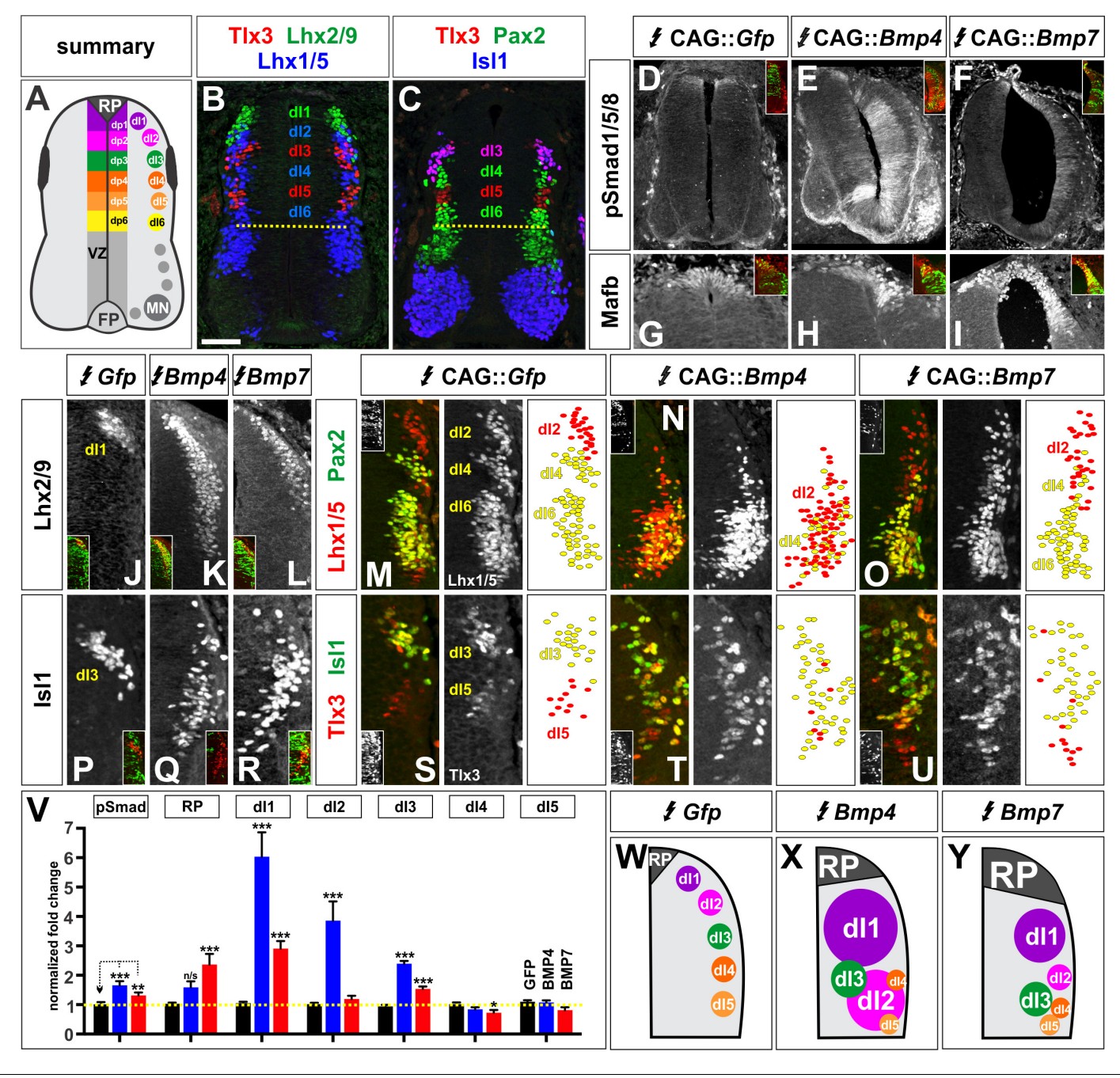

**Figure 2.** BMP4 and BMP7 direct distinct dorsal IN identities in vivo. (A–C) Summary of dorsal progenitors (dP1-6) and post-mitotic neurons (dl1-6) in the developing spinal cord. The combinatorial use of antibodies against Lhx2/9 (green), Lhx1/5 (blue), Isl1 (blue), Pax2 (green) and Tlx3 (red) permit the unambiguous identification of dl1-dl5. (D–U) Chicken spinal cords were electroporated at HH stage 15 with *Gfp* (D, G, J, M, P, S), *Bmp4* (E, H, K, N, Q, T) or *Bmp7* (F, I, L, O, R, U), under control of the CAG enhancer (*Miyazaki et al., 1989*), and incubated until HH stage 25. Thoracic transverse sections were labeled with antibodies against Tlx3 (red, **S–U**), Lhx2/9 (red, **J–L**), Lhx1/5 (red, **M–O**), Isl1 (red P-R; green, **S–U**) or Pax2 (green M-O), pSmad1/5/8 (red, **D–F**) and Mafb (red, **G–I**). (D–F) Ectopic expression of *Bmp4* (n = 52 sections from 5 embryos, p<0.0001) more effectively activates the R-Smads (Smad1/5/8) than *Bmp7* (n = 30 sections 4 embryos, p<0.005), while the expression of *Gfp* has no effect (n = 37 sections from 4 embryos, p>0.72). (G–I) Mis-expression of *Bmp4* or *Bmp7* has dramatic, but distinct, effects on dorsal cell differentiation. Specifically, the ectopic expression of *Bmp7* (I, n = 28 sections from 4 embryos, p<7.01×10$^{-5}$) resulted in consistently more Mafb$^+$ RP cells than *Bmp4* (H, n = 37 sections from 5 embryos, p<0.12), and the *Gfp* control (G, n=36 sections from 2 embryos, p>0.67). (J–L, P–R) Mis-expression of *Bmp4* however, most effectively directs cells towards the Lhx2/9$^+$ dl1 (K, n = 28 sections from 3 embryos, p<0.0001) and Isl1$^+$ dl3 fates (Q, 91 sections from 5 embryos, p<0.0001) compared to *Bmp7* (dl1, L, n = 26 sections from 3 embryos, p<0.0001; dl3, R, n = 59 sections from 5 embryos, p<5.58×10$^{-6}$). Mis-expression of *Gfp* has no effect (dl1, J, n = 46 sections

*Figure 2 continued on next page*

Figure 2 continued

from 3 embryos, p>0.47; dI3, P, n = 45 sections from 3 embryos, p>0.46). (P–R) *Bmp4* (N, n = 47 sections from 3 embryos, p<0.0001) is the only BMP sufficient to direct cells toward an Lhx1/5$^+$ dI2 identity, while *Bmp7* can suppress Pax2$^+$ dI4 fate (O, n = 24 sections from 3 embryos, p<0.01). The ectopic expression of *Gfp* has no effect (M, n = 35 sections from 3 embryos, p>0.77). Note that the presence of Bhlhb5 (*Skaggs et al., 2011*), permitted the Pax2$^+$ Lhx1/5$^-$ Bhlhb5$^-$ dI4s to be unambiguously distinguished from the Pax2$^-$ Lhx1/5$^+$ Bhlhb5$^-$ dI2s and the Pax2$^+$ Lhx1/5$^+$ Bhlhb5$^+$ dI6s (data not shown). (S–U) The mis-expression of the BMPs has no effect on the Tlx3$^+$ dI5 fate (S, GFP: n = 38 sections from 3 embryos p>0.16; BMP4: T, n = 50 sections from 4 embryos, p>0.39; BMP7: U, n = 29 sections from 3 embryos, p>0.08) fates. (V) Quantification of the fold change in cell number normalized to *Gfp* control. The probability *Bmp4* and *Bmp7* misexpression result in the same distribution of cellular activities is p<0.0002 (Fisher test). (W–Y) Summary of the cellular changes directed by BMP4 and BMP7. We found that there are there are spatial organizational changes for *Bmp4*, but not *Bmp7*, misexpression. While the RP, dI1 and dI3 populations remain in the correct spatial order with respect to each other, the dI2s both expand and change location, such that they are intermingled with/ventral to the dI3 population. Probability of similarity between control and experimental groups, *=p<0.05, **p<0.005, ***p<0.0005. Student's *t*-test or Mann-Whitney test. Scale bar: 65 µm.

DOI: https://doi.org/10.7554/eLife.30647.004

The following source data and figure supplements are available for figure 2:

**Source data 1.** BMP4 and BMP7 direct distinct dorsal IN identities in vivo (experimental data).

DOI: https://doi.org/10.7554/eLife.30647.009

**Figure supplement 1.** Other BMPs have unique activities specifying dorsal IN populations.

DOI: https://doi.org/10.7554/eLife.30647.005

**Figure supplement 2.** BMP misexpression alters the expression of the electroporated BMP.

DOI: https://doi.org/10.7554/eLife.30647.006

**Figure supplement 3.** BMP misexpression results in consistent alterations in the levels of pSmad1/5/8.

DOI: https://doi.org/10.7554/eLife.30647.007

**Figure supplement 4.** BMP misexpression can result in major morphological changes to the spinal cord.

DOI: https://doi.org/10.7554/eLife.30647.008

## BMPs direct different dorsal spinal fates in vitro

A different complement of BMPs is present in the dorsal spinal cord in mouse embryos, including BMP5, BMP6, BMP7 and GDF7 (*Figure 1—figure supplement 1*) (*Lee et al., 1998*), compared to the chicken, suggesting that the code of BMPs that specifies dorsal spinal identity may differ in different species. We addressed this question by assessing whether different BMPs can direct mouse embryonic stem cells (mESCs) towards specific dorsal IN identities. This approach also allowed us to determine the role of different BMPs on naïve tissue with minimum complication from endogenous BMP signaling. We established a directed differentiation protocol for dorsal spinal identity by modifying the protocol used to differentiate mESCs into spinal motor neurons (*Figure 3—figure supplement 1*) (*Wichterle and Peljto, 2008*). Aggregated mESCs, or embryoid bodies (EBs), were treated with 10 ng/ml of different BMP recombinant proteins, in addition to retinoic acid (RA) (*Figure 3A*). This protocol directs EBs toward a Tuj1$^+$ Hoxc8$^+$ caudal cervical/rostral thoracic spinal identity (*Figure 3B–F*) (*Philippidou and Dasen, 2013*).

Similar to our observations in vivo, we found that each BMP has a unique range of effects on the differentiation of sensory INs in vitro. Thus, BMP6 most effectively drives the expression of Msx1, a RP marker, while BMP4 most effectively directs mESCs to express Lhx2, a marker of dI1s. All BMPs tested can direct mESCs towards Isl1$^+$ dI3s (*Figure 3G*). However, BMP7 protein had only a modest effect on dorsal IN identity in this assay (*Figure 3G–H*), supporting previous analyses (*Lee et al., 1998*; *Butler and Dodd, 2003*) and suggesting that mouse BMP6 is the functional analog of chicken BMP7 in this context. Moreover, no BMP was found to direct mESCs towards the Foxd3$^+$ dI2 population, even when we tested a number of different combinations of BMPs, including a mixture of 'all' BMPs (i.e. BMP4, BMP5, BMP6, BMP7 and GDF7) (*Figure 3—figure supplement 2*). Nonetheless, these studies support the model that the BMPs have different effects directing the differentiation of the dorsal spinal IN populations, and that the specific activities of a given BMP are not necessarily evolutionarily conserved between species.

## The BMPs do not act as morphogens to direct dorsal spinal IN identity either in vivo or in vitro

Previous studies have suggested that the BMPs act as either a spatial (*Liem et al., 1997*) or temporal (*Tozer et al., 2013*) morphogen to pattern dorsal spinal identity, largely by analogy with Shh

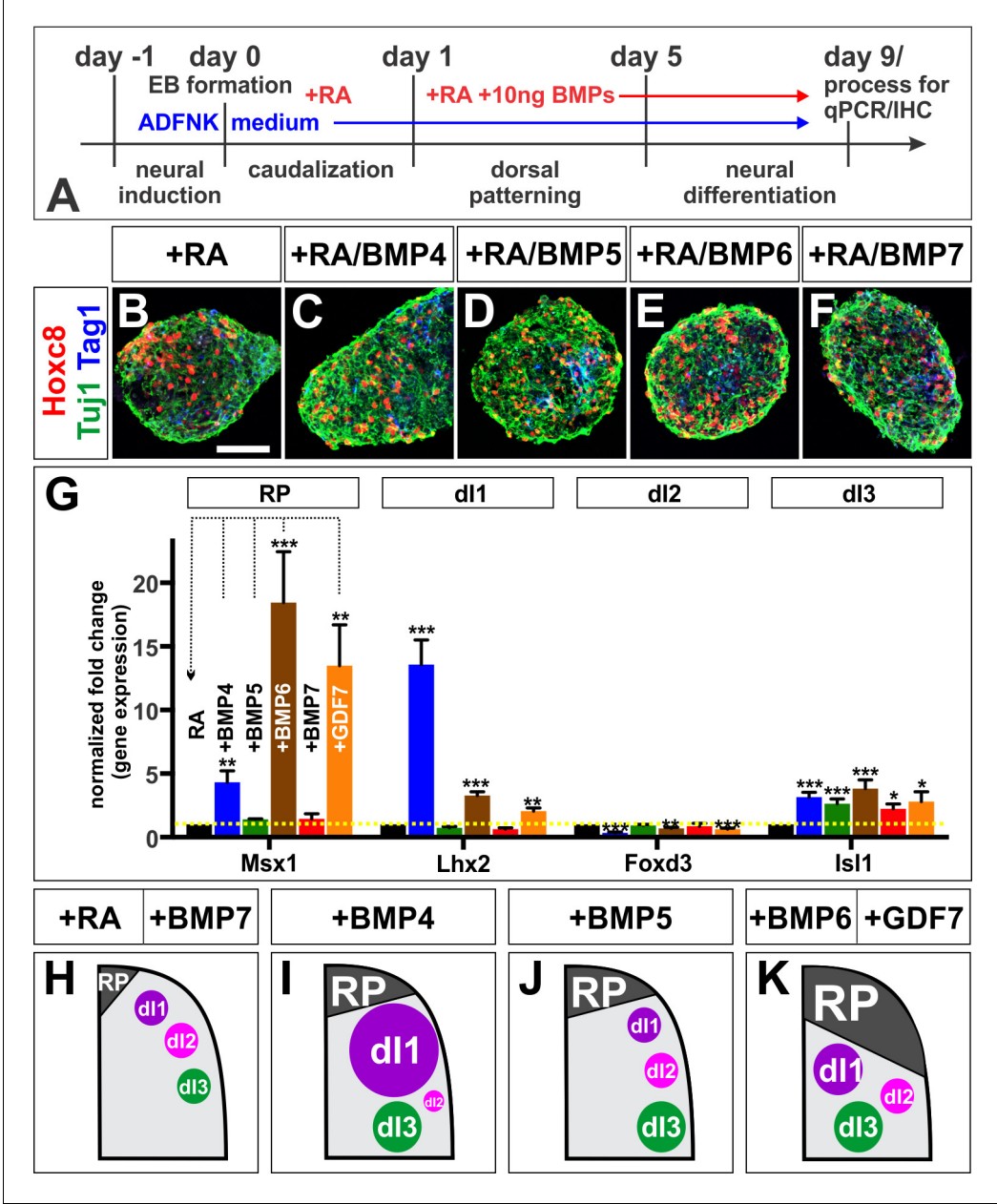

**Figure 3.** BMPs direct distinct dorsal IN identities in vitro. (**A**) Summary of the mESC differentiation protocol. (**B–F**) RA treatment ±10 ng BMPs directs EBs towards a caudal (Hoxc8[+]), neural (Tuj1[+]) and dorsal (Tag1[+]) spinal identity. (**G**) Quantification of RT-qPCR expression data from 9 day EBs, normalized to RA control (n = 5 experiments). BMP6 (n = 5 independent experiments, p<0.0005) and GDF7 (n = 5 experiments, p<0.001) are sufficient to increase Msx1 (i.e. RP) expression, whereas BMP4 (n = 5 experiments, $p<1.08\times10^{-7}$) is most effective at directing Lhx2 (dI1) expression. All BMPs tested, including BMP5 (n = 3 experiments, p<0.0002) and BMP7 (n = 2 experiments, p<0.02) elevate Isl1 (dI3) expression. No BMP tested is sufficient to elevate Foxd3 (dI2) expression. In each independent experiment, all samples were run in triplicate. (**H–K**) Summary of cell fates changes in mESC experiments. Probability of similarity between control and experimental groups, *=p < 0.05, **p<0.005, ***p<0.0005, Student's t-test. Scale bar: 65 μm.

DOI: https://doi.org/10.7554/eLife.30647.010

The following source data and figure supplements are available for figure 3:

**Source data 1.** BMPs direct distinct dorsal IN identities in vitro (experimental data).
DOI: https://doi.org/10.7554/eLife.30647.013

**Figure supplement 1.** Strategies used to establish the mESC protocol.

*Figure 3 continued on next page*

*Figure 3 continued*

DOI: https://doi.org/10.7554/eLife.30647.011

**Figure supplement 2.** Effect of combinatorial addition of BMPs on dorsal IN differentiation in vitro.

DOI: https://doi.org/10.7554/eLife.30647.012

patterning of the ventral spinal cord (*Briscoe and Novitch, 2008*). The canonical spatial morphogen model predicts that a high concentration of BMPs directs the dorsal-most spinal identities, while progressively lower concentrations of BMPs specify the dI2s, and then dI3s. In contrast, the temporal morphogen model predicts that the length of exposure to BMPs should be critical, with a given concentration of a BMP specifying first the dI3s, and then the dI1s over time. However, if BMPs rather act in a signal-specific manner, then a particular BMP ligand should direct the same range of cellular identities regardless of concentration.

We evaluated these models in both our in vivo and in vitro model systems. We altered the level of induced BMP4 signaling in vivo by *in ovo* electroporating different ratios of the CAG::*gfp* and CAG::*Bmp4* vectors into chicken embryos, keeping the total concentration of DNA constant at 500 ng/µl. We examined a concentration series consisting of 5 ng/µl CAG::*Bmp4* (i.e. a ratio of 99:1 CAG::*Gfp*: CAG::*Bmp4*), 25 ng/µl CAG::*Bmp4* (19:1), 50 ng/µl CAG::*Bmp4* (9:1) to 500 ng/µl CAG:: *Bmp4*. This concentration range resulted in a relatively linear increase in Smad1/5/8 activity: as the concentration of CAG::*Bmp4* increased, pSmad1/5/8 also increased between ~20% to~130% compared to control electroporations (*Figure 4A–C,J*). Strikingly, the different concentrations of *Bmp4* were able to direct the same range of cellular fates: all conditions resulted in increased numbers of dI1s, dI2s and dI3s (*Figure 4D–I*), however the effects became progressively larger as the concentration of BMP4 increased (*Figure 4J*). Thus, lower levels of BMP4 did not promote a more ventral-dorsal identity as predicted by the morphogen models. Rather, concentration appears to control the efficiency by which BMP4 can direct the same range of cell fates.

We more rigorously assessed the ability of BMPs to act as spatial morphogens in vitro by determining whether altering the concentration of BMP4, BMP5, BMP6 or BMP7 can direct EBs towards different dorsal spinal fates. Using a 0.15 ng/ml - 20 ng/ml concentration range, we again observed that the concentration of a given BMP controls the efficiency by which it drives its specific range of cellular identities (*Figure 4K*). Specifically, BMP4 can direct EBs to express *Lhx2* (dI1) and *Isl1* (dI3), and increasing the concentration of BMP4 drives EBs to express progressively higher levels of both *Lhx2* and *Isl1*. Similarly, increasing BMP6 levels results in progressively higher levels of *Msx1* and *Isl1* suggesting that 'high' BMP6 concurrently directs more cells to become RP and dI3s (*Figure 4K*). In no case did we observe that low versus high concentrations of BMPs can direct the ventral-dorsal versus dorsal-most cellular identities predicted by the spatial morphogen model. We further considered whether our data fit with the proposed temporal model (*Tozer et al., 2013*), which predicts that prolonged BMP signaling will direct more dorsal identities. However, we found no evidence that changing the length of time that EBs are exposed to BMPs led to progressive dorsalization (*Figure 4—figure supplement 1*). Thus, our studies support a signal-specific model where each BMP ligand can direct progenitors towards a unique range of dorsal spinal identities.

## BMP4 and BMP7 have differential effects on dorsal progenitor identity in vivo

We next examined the mechanistic basis by which BMP4 and BMP7 direct distinct dorsal spinal identities in chicken embryos. We first assessed how broadly *Bmp4* or *Bmp7* misexpression affects neural development by examining their effect on Sox2[+] progenitors (*Bylund et al., 2003*) and p27/Kip1[+] differentiated neurons (*Novitch et al., 2001*) (*Figure 5A–C*). Neither *Bmp4* nor *Bmp7* misexpression changed the ratio of Sox2[+] progenitors to p27/Kip1[+] neurons (*Figure 5A–C,G*), although the spinal cord was generally more elongated after *Bmp4* expression (*Figure 5B,G*). However, *Bmp4* misexpression significantly diminished ($p < 1 \times 10^{-12}$) the intensity of Sox2 in dorsal progenitors (bracket, *Figure 5B,G*), suggesting that progenitor identity is being eroded, perhaps because the cells are preparing to differentiate.

We then looked more specifically at the ability of BMP4 and BMP7 to mediate dorsal progenitor identity, at HH stage 20, a day after electroporation. Ectopic expression of either *Bmp4* or *Bmp7*

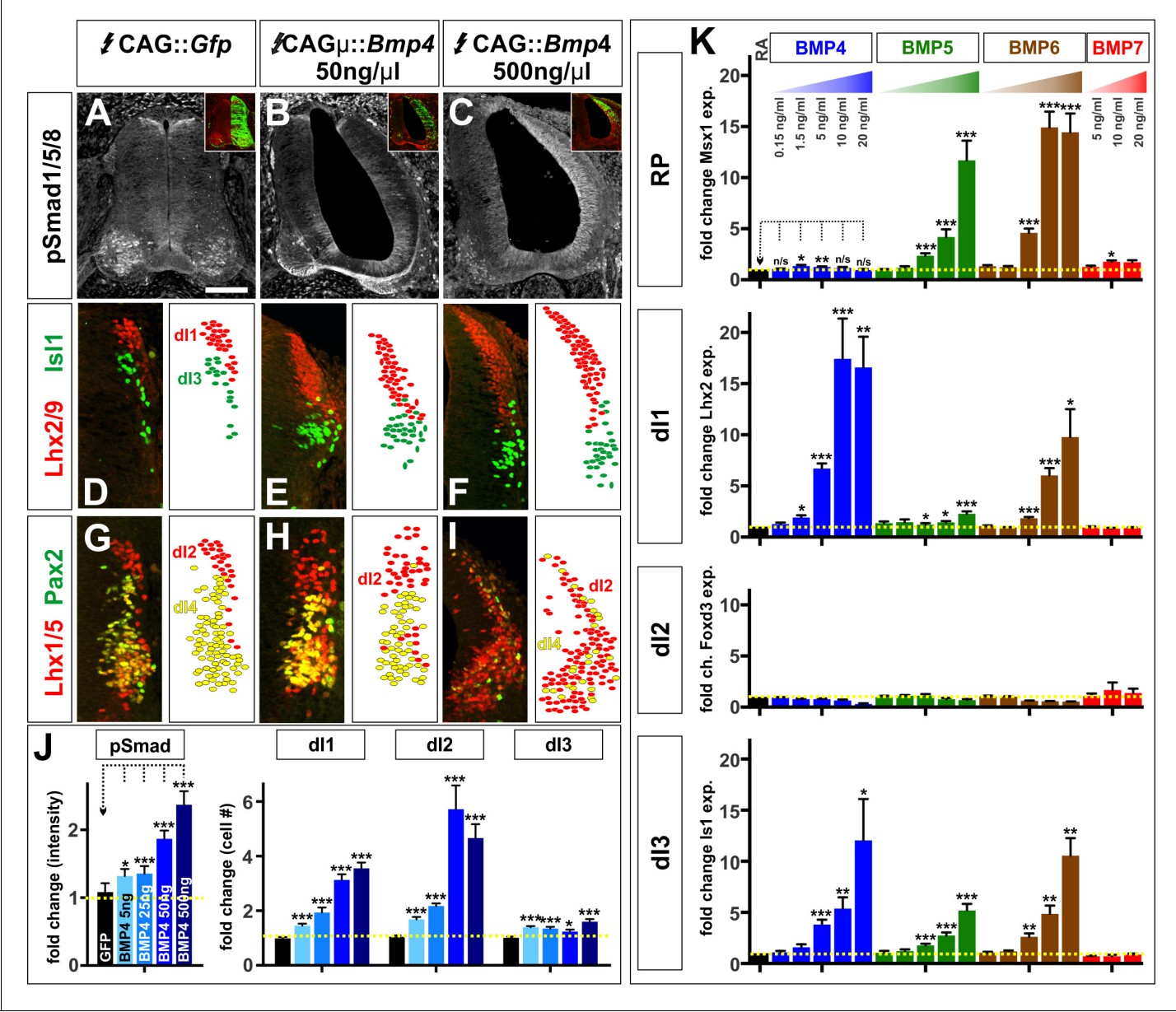

**Figure 4.** The BMPs do not act as morphogens either in vivo or in vitro. (A–I) Chicken spinal cords were electroporated at HH stage 15 with *Gfp* (A, D, G), [low] *Bmp4* (B, E, H) or [high] *Bmp4* (C, F, I) under the control of the CAG enhancer, and incubated until HH stage 25. Thoracic transverse sections were labeled with antibodies against pSmad1/5/8 (red, A–C), Lhx2/9 (red, D–F), Isl (green, D–F), Lhx1/5 (red, G–I) and Pax2 (green, G–I). (A–C, J) R-Smad activation increases over a range of ~20% to~130% as the level of *Bmp4* misexpression increases. 5 ng/µl CAG::*Bmp4*, n = 27 sections from 3 embryos, p<0.008 similar to control; 25 ng/ul CAG::*Bmp4*, n = 14 sections from 2 embryos, p<4.4×10$^{-5}$; 50 ng/ul CAG::*Bmp4*, n = 23 sections from 3 embryos, p<2.4×10$^{-8}$; 500 ng/ul CAG::*Bmp4*, n = 19 sections from 3 embryos, p<3.6×10$^{-7}$; *Gfp* control: n = 27 sections from 3 embryos, p>0.47. (D–I, J) The concentration of BMP4 determines the efficiency of dl1-3 specification in chicken embryos. Thus, high levels of *Bmp4* expression direct more Lhx2/9$^+$ dl1s (F, n = 28 sections from 3 embryos, p<1.8×10$^{-16}$), Lhx1/5$^+$ dl2s (I, n = 22 sections from 3 embryos, p<7.6×10$^{-11}$) and Isl1$^+$ dl3s (F, n = 28 sections from 3 embryos, p<6.3×10$^{-7}$) compared to lower concentrations of *Bmp4* expression (E, Lhx2/9: n = 29 sections from 3 embryos, p<7.6×10$^{-14}$; Isl1: n = 29 sections from 3 embryos, p<0.006; H, Lhx1/5: n = 26 sections from 3 embryos, p<5.1×10$^{-7}$). Expression of *Gfp* had no effect (Lhx2/9: D, n = 28 sections from 3 embryos, p>0.88; Isl1: D, n = 29 sections from 3 embryos, p>0.89; Lhx1/5: G, n = 29 sections from 3 embryos, p>0.28). (K) Similarly, increasing the concentration of a given BMP improves its ability to direct mESCs towards a specific dorsal fate, as measured by increased gene expression. BMP4: n = 4 independent experiments; BMP5: n = 3 experiments; BMP6: n = 2 experiments; BMP7: n = 2 experiments. Samples were run in triplicate within each experiment. Probability of similarity between control and experimental groups, *p<0.05, **p<0.005, ***p<0.0005, Student's *t*-test or Mann Whitney test. Scale bar: 80 µm.

DOI: https://doi.org/10.7554/eLife.30647.014

*Figure 4 continued on next page*

*Figure 4 continued*

The following source data and figure supplement are available for figure 4:

**Source data 1.** The BMPs do not act as morphogens either in vivo or in vitro (experimental data).

DOI: https://doi.org/10.7554/eLife.30647.016

**Figure supplement 1.** Temporal effect of the BMPs on dorsal IN identity in vitro.

DOI: https://doi.org/10.7554/eLife.30647.015

increased the length of the Pax3$^+$ dorsal progenitor domain by ~60% (*Figure 5D–G*). However, the two BMPs had markedly different effects on individual progenitor identities within this domain. *Bmp7* had the most dramatic effect on the area of the *Atoh1$^+$* dp1 domain (*Figure 5J*), with a more modest effect on the *Ascl1$^+$* dp3-5 domain (*Figure 5S*) and no effect on the *Ngn1$^+$* dp2 domain (*Figure 5P*). In contrast, *Bmp4* can increase the area of both the *Atoh1$^+$* dp1 and *Ascl1$^+$* dp3-5 domains (*Figure 5I,R*), but markedly decreases the size of the *Ngn1$^+$* dp2 domain (*Figure 5O*). Taken together, these results suggest that the ability of BMP4 and BMP7 to instruct specific ranges of dorsal spinal fates stems from their differential effects on dorsal progenitors.

## BMPs differentially affect cell cycle dynamics in vivo and in vitro

Based on previous studies showing that BMPs can regulate proliferation and neurogenesis (*Shou et al., 2000*; *Shi and Liu, 2011*), we hypothesized that the differential effects of BMP4 and BMP7 on dorsal progenitors arise from BMPs having distinct effects on progression through the cell cycle. A progenitor-patterning factor might speed up cell cycle transitions, whereas a differentiation factor might promote cell cycle exit. Using BrdU as a marker for cells in synthesis (S) phase (*Djordjevic and Szybalski, 1960*), we found that misexpression of *Bmp7*, but not *Bmp4*, can decrease the number of BrdU$^+$ cells in the dP1 and dP2 domains in vivo (*Figure 6A–I,M*). In contrast, ectopic expression of both *Bmp4* and *Bmp7* increases the number of pHistoneH3$^+$ cells i.e. cells in mitosis (M)-phase (*Goto et al., 1999*) (*Figure 6J–L,N*). Thus, BMP4 and BMP7 have different effects on the cell cycle in the chicken spinal cord: BMP7 can simultaneously decrease the number of cells in S-phase and increase the number of cells in M-phase, consistent with shortening the duration of the cell cycle. In contrast, BMP4 specifically increases the number of dividing cells only.

We further evaluated the effect of BMPs on cell cycle dynamics, by assessing these cell cycle markers in mESCs treated with BMP4, BMP5, BMP6, BMP7 or GDF7. We found that only BMP4 and BMP6 were able to affect these markers as neural progenitors progress through the cell cycle (*Figure 7B,D,S*). Strikingly, treatment with BMP4 resulted an ~2 fold increase in the number of cells in S-phase (*Figure 7H,S*) and M-phase (*Figure 7N,T*). In contrast, BMP6 had a more modest effect increasing the number of cells in S-phase only (*Figure 7J,P,S,T*). Thus, our in vitro studies are consistent with those in vivo in that different BMPs have distinct effects modulating the cell cycle. They also support the hypothesis that mouse BMP6 is the functional analog of chicken BMP7.

Taken together, the activities of the BMPs in vivo are consistent with a model where both BMP7 and BMP4 drive progenitor patterning, while BMP4 directly promotes neuronal differentiation (*Figure 6O*). We assessed this model by examining the timing by which *Bmp4* or *Bmp7* expression can direct dI1 differentiation. Both *Bmps* can increase the size of the dP1 domain, 24 hr post-electroporation (*Figure 5T*), and the numbers of dI1s, 48 hr post-electroporation, with BMP4 exerting the largest effect (*Figure 2V*). However, only the misexpression of *Bmp4* results in increased dI1 differentiation within 24 hr after electroporation (*Figure 5L,T*). Thus, only BMP4 appears to instruct endogenous progenitors to differentiate as dI1s, while BMP7 increases the number of dI1s as a secondary consequence of promoting increased dP1 patterning. Collectively these studies support the hypothesis that the BMPs have signal-specific effects on progenitor patterning and differentiation, which may be mediated by regulating cell cycle dynamics. Supporting this model, the misexpression of either *Bmp5* or *Gdf7 in vivo* also uniquely affects the patterning of dorsal progenitors and their progression through the cell cycle (*Figure 6—figure supplement 1*).

## Different BMP type I receptors mediate the activities of the BMPs

Finally, we assessed whether the distinct effects of these BMPs on dorsal spinal neurogenesis are a consequence of differential signaling through distinct receptor complexes. The BMPs bind to a

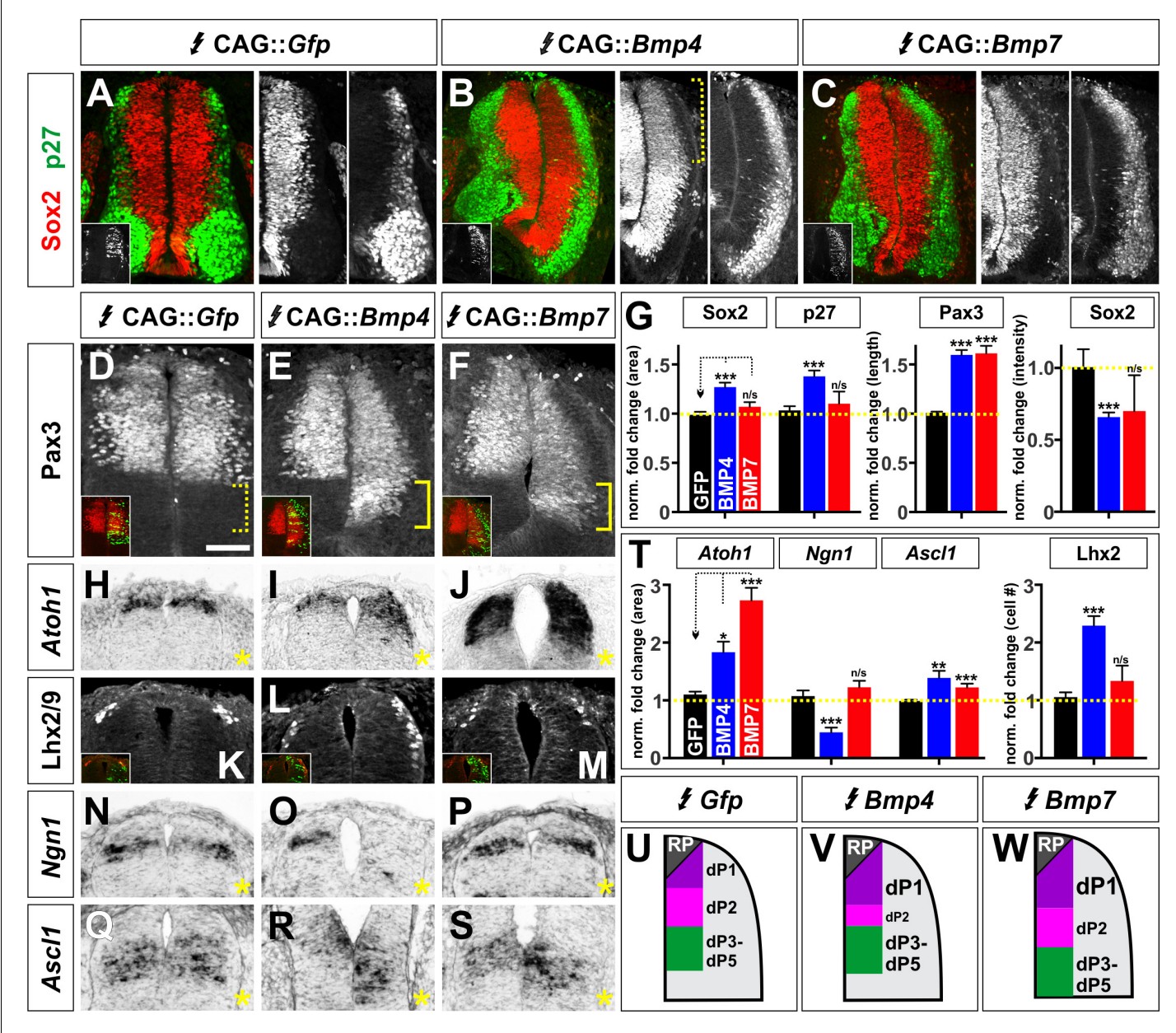

**Figure 5.** BMP4 and BMP7 can drive distinct dorsal progenitor identities in vivo. (A–S) Chicken spinal cords were electroporated at HH stage 15 with *Gfp* (A, D, H, K, N, Q), *Bmp4* (B, E, I, L, O, R) or *Bmp7* (C, F, J, M, P, S) under the control of the CAG enhancer, and incubated until HH stage 25 (A–F) or 20 (J–S). Thoracic transverse sections were labeled with antibodies against Sox2 (red, A–C), p27 (green, A–C) or Lhx2/9 (K–M) or processed for in situ hybridization (H–J, N–S). Note that *Bmp* misexpression affects progenitor identity on both sides of the spinal cord, thus it is critical to compare the experimental manipulations to the GFP electroporation control. (A–C, G) The ratio of progenitors to neurons was not affected by the misexpression of *Gfp* (n = 40 sections from 3 embryos), *Bmp4* (n = 47 sections from 4 embryo) or *Bmp7* (n = 27 sections from 3 embryos), although the spinal cord was generally elongated in *Bmp4* electroporated embryos (A–C, G). *Bmp4* misexpression also significantly diminished Sox2 intensity by >30% in the dorsal spinal cord (bracket, B, p<9.22×10$^{-13}$). (D–G) Misexpression of *Bmp4* (n = 50 sections from 5 embryos, p<5.2×10$^{-19}$) and *Bmp7* (n = 29 sections from 3 embryos, p<9.6×10$^{-17}$) significantly increases the size of the Pax3$^+$ domain compared to *Gfp* control (n = 44 sections from 3 embryos, p>0.95). (H–J) The misexpression of *Bmp4* (n = 54 sections from 4 embryos, p<0.006) and *Bmp7* (n = 25 sections from 3 embryos, p<3.0×10$^{-14}$) results in a 2–3 fold increase in the area of the *Atoh1*$^+$ +1 domain compared to *Gfp* control (n = 42 sections from 6 embryos, p>0.16). (K–M) 24 hr post-electroporation, *Bmp4* expression (n = 25 sections from 3 embryos, p<2.35×10$^{-10}$) increases the number of Lhx2/9$^+$ dI1s while the expression of *Gfp* (n = 34 sections from 3 embryos, p>0.6) or *Bmp7* (n = 25 sections from 3 embryos, p>0.74) has no significant effect. (N–P) *Bmp4* (n = 44 sections from 3 embryos, p<0.0001) decreases the size of the *Ngn1*$^+$ dp2 domain while neither *Gfp* (n = 176 sections from 8 embryos, p>0.23) or *Bmp7* (n = 41 sections from 3 embryos, p>0.62) expression has a significant effect. (Q–S) The misexpression of both *Bmp4* (n = 48 sections from 3 embryos, p<0.003) and *Bmp7* (n = 45 sections from 3 embryos, p<4.2×10$^{-5}$) significantly increase the *Ascl1*$^+$ dP3-5 populations compared to *Gfp* control (n = 124 sections from 8

*Figure 5 continued on next page*

*Figure 5 continued*

embryos, p>0.73). (T) Quantification of the fold change in progenitor domain area or cell number normalized to the GFP control. (U–W) Summary of changes to progenitor identity directed by BMP4 or BMP7. Probability of similarity between control and experimental groups, *p<0.05, **p<0.005, ***p<0.0005, Student's *t*-test or Mann-Whitney test. Scale bar: 50 μm.

DOI: https://doi.org/10.7554/eLife.30647.017

The following source data is available for figure 5:

**Source data 1.** BMP4 and BMP7 can drive distinct dorsal progenitor identities in vivo (experimental data).

DOI: https://doi.org/10.7554/eLife.30647.018

complex of type I and type II BMP receptors (Bmprs). Previous studies have suggested that the BMP type I receptors, BmprIa (Alk3) and BmprIb (Alk6), are expressed in progenitors and neurons respectively and mediate different aspects of embryonic development (*Yamauchi et al., 2008*; *Panchision et al., 2001*). We confirmed that the type I Bmprs have similarly distinct distributions in the developing chicken spinal cord. *BmprIa* is expressed in progenitors prior to HH stage 18 and remains restricted to the VZ (*Figure 8A–C*). In contrast, *BmprIb* expression starts by HH stage 21 and continues to be present in both progenitor and neuronal domains by HH stage 24 (*Figure 8D–F*).

To assess the effects of BmprIa and BmprIb on dorsal IN identity, we electroporated dominant negative forms of the type I Bmprs (*Lim et al., 2005*) into chicken embryos. Both dn*BmprIa* and dn*BmprIb* were able to block the formation of Lhx2/9$^+$ dI1s and Isl1$^+$ dI3s (*Figure 8G–I*). However dn*BmprIb* was significantly (p<1×10$^{-5}$) more effective at reducing the number of dI1s (*Figure 8M*) and was the only BmprI that decreases the number of Lhx1/5$^+$ Pax2$^-$ dI2s (*Figure 8L,M*). Thus, dnBmpIb appears to block the dorsal IN populations specified by BMP4, whereas dnBmprIa may be blocking the activities mediated by BMP7. We further assessed this possibility in both our in vivo and in vitro model systems. We performed a rescue experiment in vivo, to determine whether it was possible to overcome the receptor blockade with co-electroporation of either BMP4 or BMP7. Supporting our hypothesis, we found that increasing the levels of BMP4 are able to fully rescue the effects of dnBmprIa, but not dnBmprIb (*Figure 8N*), i.e. BMP4 is able to circumvent blocking BmprIa, possibly by activating endogenous BmprIb, but BMP4 is not able to circumvent the effect of inhibiting BmprIb. The results with BMP7 are less clear-cut, however the specificity does appear to be reversed: high levels of BMP7 are only able to fully rescue the effects of dnBmprIb.

We also assessed whether the differentiation activity of BMP4 or BMP6, which has comparable activities to chicken BMP7 for mESCs, could be blocked by inhibiting different type I receptors in our mESC in vitro assay. We used two BMP receptor inhibitors, dorsamorphin and LDN-193189 (*Cuny et al., 2008*), which inhibit BmprIa at low concentrations, but only block BmprIb at 5- to 10-fold higher concentrations (*Yu et al., 2008a*, *2008b*). Low concentrations of both inhibitors were effective at blocking the activities of BMP6 directing either Msx1 or Lhx2 expression (*Figure 8O,P*). In contrast, it took 5 to 10-fold more of either inhibitor to block the activities of BMP4 (*Figure 8P*). These studies provide strong support for a model in which BmprIa mediates the activities of BMP6, while BmprIb mediates BMP4 signaling. Together, our in vivo and in vitro analyses demonstrate that different BMP ligands to act through distinct type I receptors to mediate their specific differentiation activities.

## Discussion

### BMPs have distinct roles directing dorsal spinal fates

We have methodically tested the ability of the dorsal complement of BMPs to direct dorsal spinal fates. The identity of BMPs present in the dorsal-most spinal cord differs between species. We have confirmed that BMP4 and BMP7, but not BMP5, BMP6 and GDF7, are present in the chicken dorsal spinal cord, whereas a larger complement of BMPs, including BMP5, BMP6, BMP7 and GDF7, but not BMP4, are present in the early mouse spinal cord. Using in vivo and in vitro approaches, we have shown that each BMP has a unique, species-specific range of cellular activities directing dorsal fates. We propose these activities result in a 'mix and match' code of BMP signaling specifying four classes of dorsal cell types (*Figure 8Q–T*). In this model, BMP6 (mouse) and BMP7 (chicken) are the most

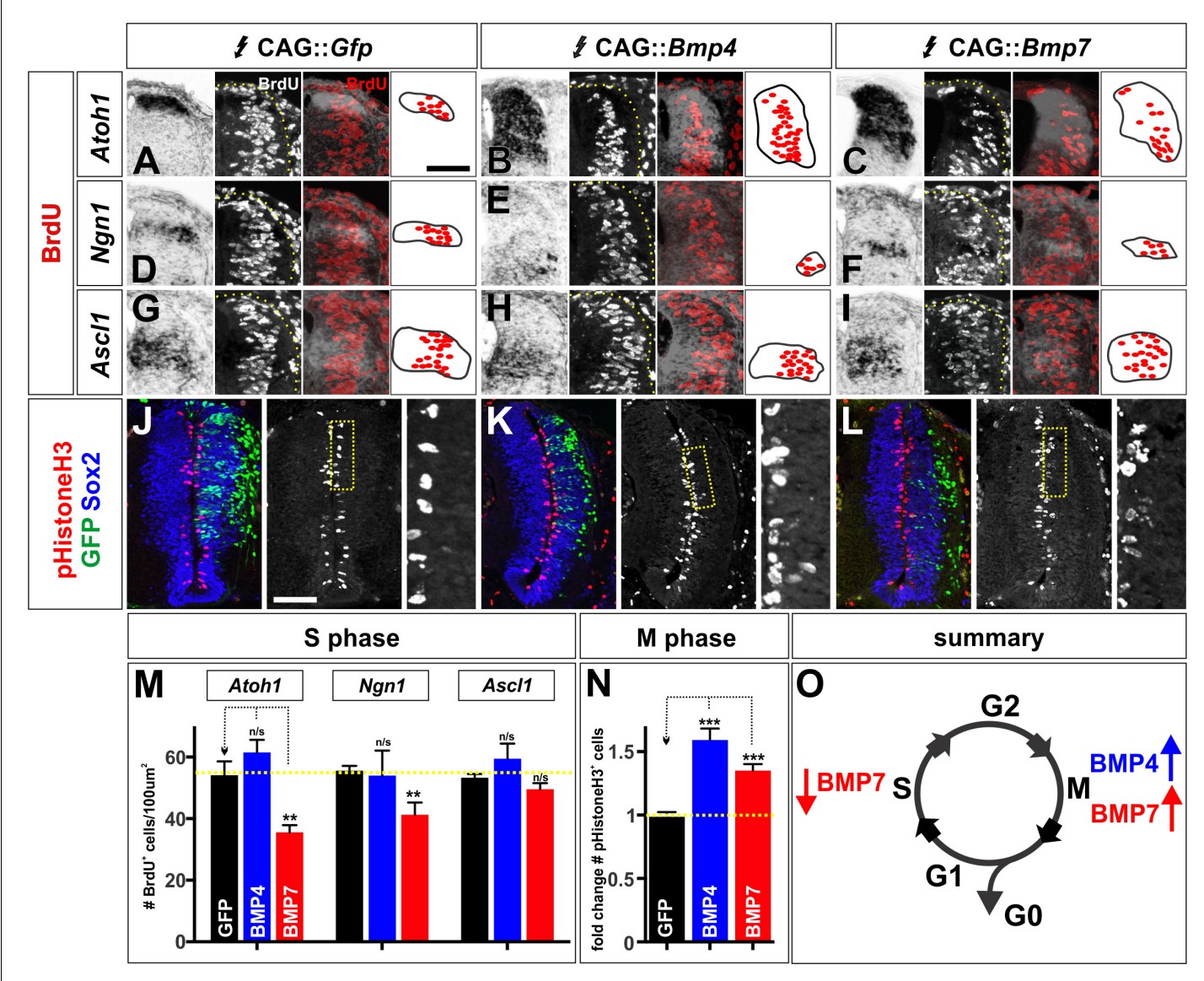

**Figure 6.** The BMPs can differentially regulate the cell cycle in vivo. (A–L) Chicken spinal cords were electroporated at HH stage 15 with *Gfp* (A, D, G, J), *Bmp4* (B, E, H, K) or *Bmp7* (C, F, I, L) under the control of the CAG enhancer and incubated until HH stage 20. Thoracic transverse sections were labeled with antibodies against pHistoneH3 (red, J–L) and Sox2 (blue, J–L). In situ hybridization was performed in combination with BrdU labeling (white/red, A–I). (A–I) *Bmp7* (Atoh1: n = 25 sections from 5 embryos, p<0.005; Ngn1: n = 41 sections from 5 embryos, p<0.001; Ascl1: n = 45 from 5 embryos, p>0.13) misexpression decreases the number of S-phase BrdU$^+$ cells per 100 um$^2$ in the dP1-dP2 domains while neither *Gfp* (Atoh1: n = 42 sections from 6 embryos, p>0.96; Ngn1: n = 176 sections from 8 embryos, p>0.54; Ascl1: n = 124 sections from 8 embryos, p>0.73) or *Bmp4* expression (Atoh1: n = 54 sections from 4 embryos, p>0.16; Ngn1: n = 44 sections from 3 embryos, p>0.28; Ascl1: n = 48 sections from 3 embryos, p>0.25) has no effect. (J–L) Both *Bmp4* (n = 47 sections from 3 embryos, p<2.2×10$^{-9}$) and *Bmp7* misexpression (n = 77 sections from 3 embryos, p<0.0001) can increase the number of pHistoneH3$^+$ M-phase cells compared to *Gfp* control (n = 60 sections from 3 embryos, p>0.81). (O) The combined effect of the BMPs on the cell cycle may permit BMP7 to most effectively promote progenitor patterning while BMP4 most effectively directs neural differentiation. Probability of similarity between control and experimental groups, *p<0.05, **p<0.005, ***p<0.0005 Student's *t*-test or Mann-Whitney test. Scale bar: 70 µm.

DOI: https://doi.org/10.7554/eLife.30647.019

The following source data and figure supplement are available for figure 6:

**Source data 1.** The BMPs can differentially regulate the cell cycle in vivo (experimental data).
DOI: https://doi.org/10.7554/eLife.30647.021
**Figure supplement 1.** BMP5 and GDF7 have unique effects on cell cycle in vivo.
DOI: https://doi.org/10.7554/eLife.30647.020

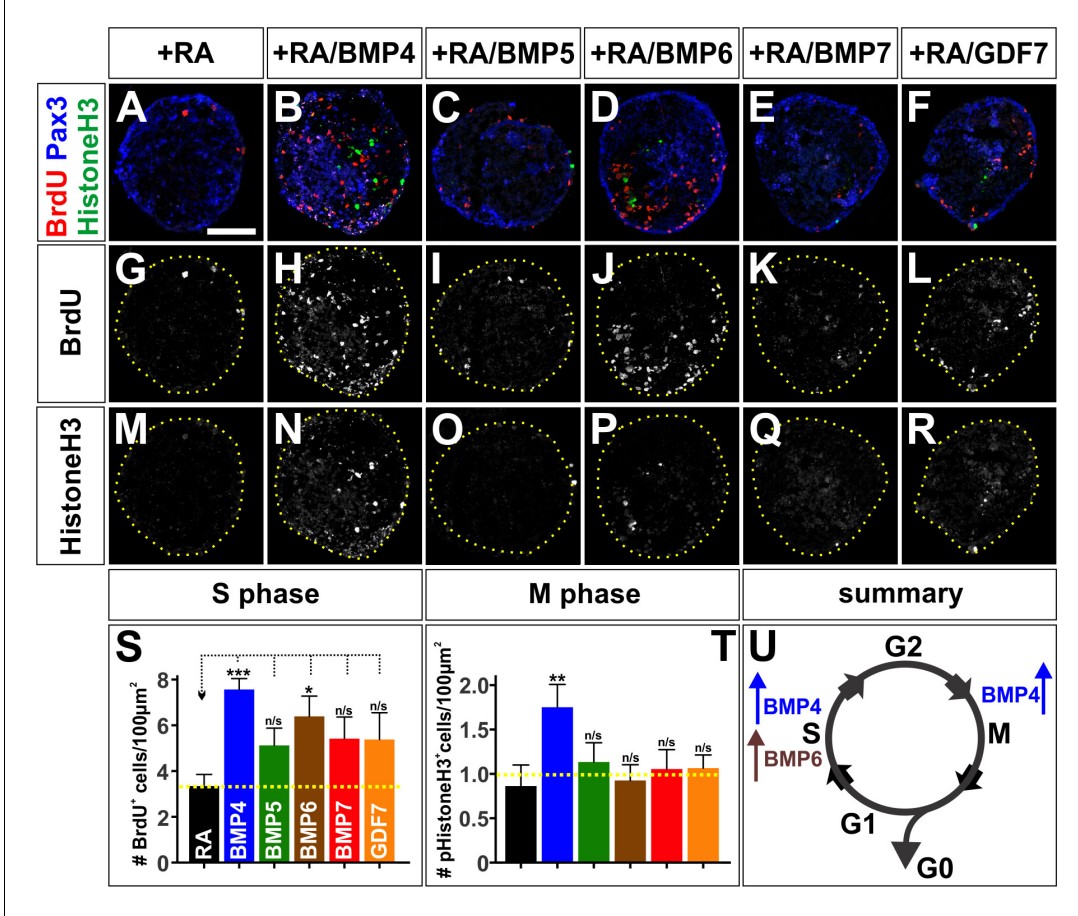

**Figure 7.** The BMPs differentially regulate the cell cycle in vitro. (**A–R**) Treatment of mEBs with RA ±10 ng BMPs directs them all towards Pax3[+] dorsal spinal fates. However, different BMPs have distinct effects promoting different phases of the cell cycle. BMP5 (**C**, n = 32 sections 3 independent experiments; **I**: BrdU, p>0.26; **O**: pHistoneH3, p>0.24), BMP7 (**E**, n = 27 sections 3 independent experiments; **K**: BrdU, p>0.09, **Q**: pHistoneH3, p>0.85), and GDF7 (**F**, n = 40 sections 3 independent experiments; **L**: Brdu, p>0.22, **R**: pHistoneH3, p>0.24) had no effect either S phase or M phase, compared to RA control (**A, G, M**, n = 21 sections 3 independent experiments). In contrast, BMP4 both increased the number of S-phase cells (**B**, n = 50 sections 3 independent experiments; **H**: BrdU, p<1.15×10$^{-6}$) and the number of M-phase cells (**B, N**: pHistoneH3, p<0.04), while BMP6 only increased the number of S-phase cells (**D**, n = 32 sections 3 independent experiments; **J**: BrdU, p<0.01) with no effect on the number of cells in mitosis (**D, R**: pHistoneH3, p<0.83). (**S–T**) Quantification of the number of BrdU $^{+}$(**S**) and pHistoneH3$^{+}$ (**T**) cells per 100 um$^2$. BMP treatment had no effect on EB size (data not shown). (**U**) Summary of the effects of BMP6 and BMP7 on cell cycle in vitro. BMP4 simultaneously increases the number of cells in S-phase and M-phase, consistent with prolonging cell cycle length. In contrast, BMP6 specifically increases the number of cells in S-phase with no effect on M-phase, possibly promoting proliferation. Probability of similarity between control and experimental groups, *=p < 0.05, **p<0.005, ***p<0.0005, Student's t-test or Mann-Whitney test. Scale bar: 80 μm.

DOI: https://doi.org/10.7554/eLife.30647.022

The following source data is available for figure 7:

**Source data 1.** The BMPs differentially regulate the cell cycle in vitro (experimental data).
DOI: https://doi.org/10.7554/eLife.30647.023

effective at directing RP identity through the Bmprla receptor (mouse) (*Figure 8Q*). Both BMP4 and BMP7 can promote dP1 patterning through Bmprla or Bmprlb (chicken), but only BMP4 directs progenitors to differentiate as dl1s through Bmprlb (mouse and chicken) (*Figure 8R*). BMP4 also specifically directs dI2 differentiation in chicken, thereby depleting the pool of dP2s (*Figure 8S*). All BMPs tested in both species, including BMP4, BMP5, BMP6 and BMP7, can act though either Bmprla or Bmprlb to promote the dl3 fate (*Figure 8T*). Our studies also show that the BMPs do not direct the dl4-dl6 fates (*Figure 2V* and data not shown). While BMP7 can promote the dl1 and dl3 fate, we find no evidence that it also directs the dl5 fate as previously suggested (*Le Dréau et al., 2012*).

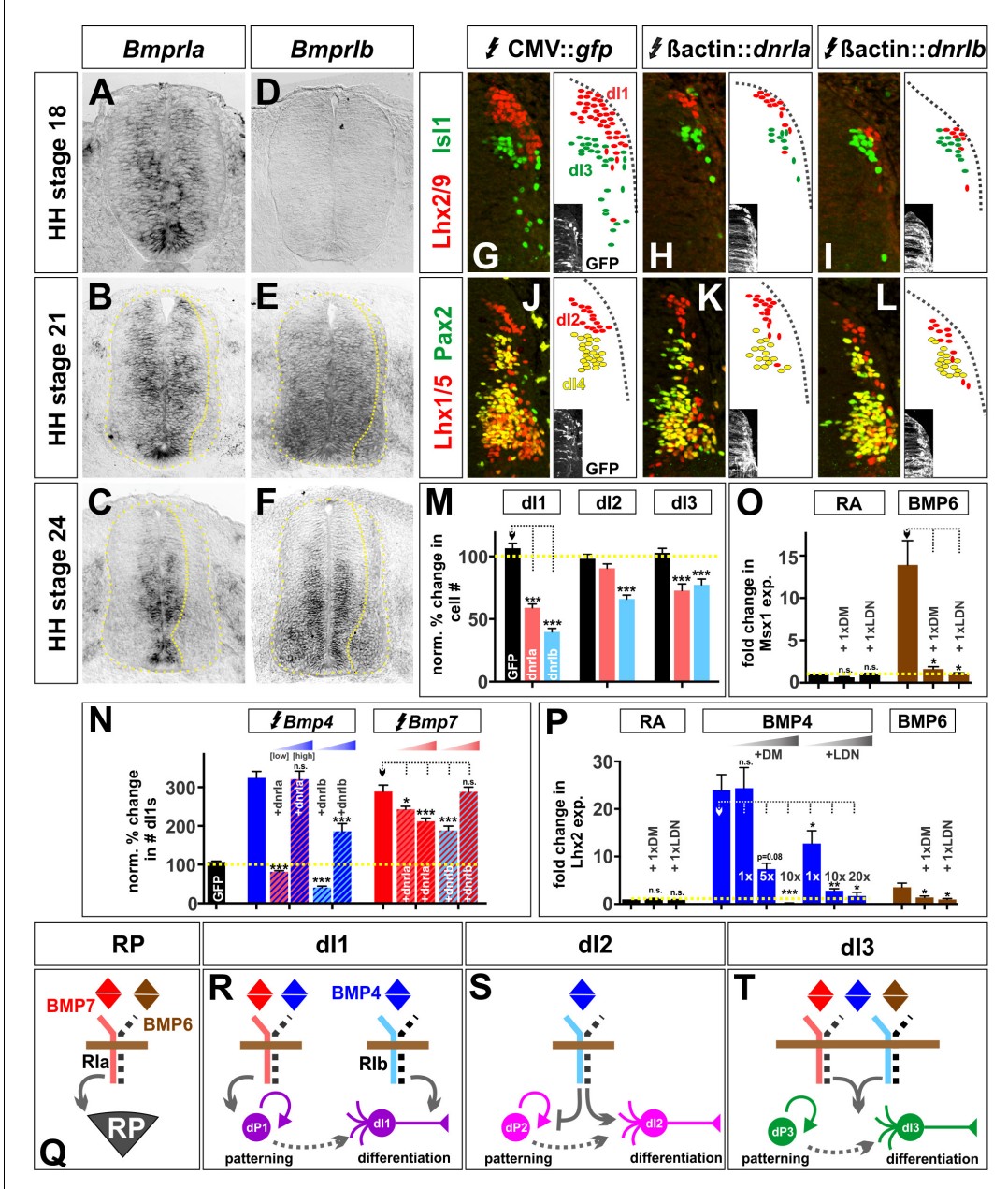

**Figure 8.** BMP4 and BMP7 mediate their diverse activities through different type I Bmp receptors both in vivo and in vitro. (A–C) *Bmprla* is expressed in spinal progenitors during neurogenesis. (D–F) At later stages, *Bmprlb* is also expressed by both progenitors and post-mitotic neurons. (G–L) Chicken spinal cords were ubiquitously electroporated with *Gfp* (G, J), *dnBmprla* (H, K) or *dnBmprlb* (I, L) at HH stage 15 and incubated until HH stage 25. Thoracic transverse sections were labeled with antibodies against Lhx2/9 (red, G–I), Isl1 (green, G–I), Lhx1/5 (red, J–L) and Pax2 (green, J–L). (G–M) The misexpression of dominant negative (dn) *Bmprlb* decreases the Lhx2/9$^+$ dI1 (I, n = 81 sections from 4 embryos, p<1.73×10$^{-24}$), Lhx1/5$^+$ dI2 (L, n = 58 sections from 4 embryos, p<2.3×10$^{-10}$) and Isl1$^+$ dI3 (I, n = 59 sections from 4 embryos, p<0.0003) populations compared to *Gfp* control (G, Lhx2/9: n = 45 sections from 5 embryos, p>0.23; J, Lhx1/5: n = 44 sections from 5 embryos, p>0.73; G, Isl1: n = 46 sections from 5 embryos, p>0.55). In contrast, the presence of dn*Bmprla* decreases the number of dI1s (H, n = 55 sections from 3 embryos, p<1.75×10$^{-13}$) and dI3s (H, n = 35 sections from 3 embryos, p<2.37×10$^{-5}$), but not the dI2s (K, n = 53 sections from 3 embryos, p>0.07). (N) Increasing the levels of *Bmp4* expression is sufficient to rescue the dn*Bmprla*, but not dn*Bmprlb*, phenotype (BMP4: n = 40 sections from 3 embryos; BMP4 low + dnIa: n = 51 sections from 3 embryos, p<3.93×10$^{-28}$; BMP4 high + dnIa: n = 32 sections from 5 embryos, p>0.88; BMP4 low + dnIb: n = 43 sections from 3 embryos, p<7.17×10$^{-29}$; BMP4 high + dnIb: n = 42 sections from 6 embryos, p<8.44×10$^{-9}$), while high levels of *Bmp7* are most effective at rescuing the dn*Bmprlb*, rather than dn*Bmprla*, phenotype (BMP7: n = 42 sections from 3 embryos; BMP7 low + dnIa: n = 57 sections from 3 embryos, p<0.008; BMP7 high + dnIa: n = 31 sections from 4 embryos, p<0.0004; BMP7 low + dnIb: n = 58 sections from 3 embryos, p<0.0001; BMP7 high + dnIb: n = 23 sections from 4 embryos, p>0.98). (O–P) Low concentrations of either dorsomorphin (DM) or LDN-193189 (LDN) are sufficient to block the activity of BMP6 in vitro, consistent with

*Figure 8 continued on next page*

*Figure 8 continued*

the model that BMP6 acts through BmprIa. In contrast, much higher concentrations of either DM or LDN are required to block BMP4-directed differentiation, suggesting BMP4 acts through BmprIb (n = 4 independent experiments). (Q–T) Models for the specification of the RP, dI1s, dI2s and dI3s. Probability of similarity between control and experimental groups, *=p<0.05, **p<0.005, ***p<0.0005 Student's *t*-test or Mann-Whitney test. Scale bar: 65 μm.

DOI: https://doi.org/10.7554/eLife.30647.024

The following source data is available for figure 8:

**Source data 1.** BMP4 and BMP7 mediate their diverse activities through different type I Bmp receptors both in vivo and in vitro (experimental data).
DOI: https://doi.org/10.7554/eLife.30647.025

Rather, these studies are consistent with our earlier finding that BMP signaling must be suppressed for the ventral-dorsal fates to be realized (*Hazen et al., 2011*).

While BMPs clearly have distinct activities encoding dorsal fates, there is generally no single function for any particular BMP. Rather, each BMP can specify multiple dorsal cell types, while markedly differing in the effectiveness by which they induce a given fate. The range of cellular activities specified by BMPs appears to be evolutionarily conserved, but the specific identity of the BMP directing those cellular identities is not. For example, BMP7 and BMP6 have similar activities in chicken and mouse respectively, most prominently promoting RP formation. Similarly, BMP4 directs the dI2 fate in chicken, but mouse BMP4 does not. Thus the code of BMP signaling that directs dIN formation appears to be species specific. It should also be noted that our in vivo manipulations of BMP signaling were performed in a background of endogenous BMP signaling. While we found that misexpression of *Bmps* tended, if anything, to reduce the expression of the endogenous BMPs (chevrons, *Figure 2—figure supplement 2C*) suggesting that the ectopic BMP may overwhelm endogenous BMP signaling, it remains a caveat that we may be examining the additive effects of BMP signaling in the chicken experiments.

The overlapping activities of the BMPs may explain why it has been challenging to dissect the signal-specific differences between the BMPs using loss-of-function approaches. For example, both BMP4 and BMP7 are required to specify the dI1s in chicken (*Le Dréau et al., 2012*). However, our sufficiency experiments have shown that BMP4 more effectively directs the dI1 fate than BMP7 (*Figure 2V*) and we have identified a mechanism to explain the difference between their activities (*Figure 8R*). In our model, both BMP7 and BMP4 can induce progenitors to become dP1s, but only BMP4 can direct progenitors to differentiate early as dI1s. Thus the larger increase in the dI1 population observed after *Bmp4* electroporation may stem from BMP4's ability to promote both dP1 patterning and dI1 differentiation (*Figure 2V*). In contrast, the smaller increase in the number of dI1s seen after *Bmp7* electroporation may result from the ectopic dP1s being directed to differentiate by endogenous BMP4.

We did identify two activities unique to BMPs: BMP7 drives the formation of the RP, and BMP4 can drive some progenitors, including the dP2s, to directly differentiate as dI2s in chicken. This latter activity is profound, leading to the depletion of dP2s (*Figure 5O*) and the formation of a large mass of Lhx1/5$^+$ dI2-like cells (*Figure 2—figure supplement 4D–F*). The dI2s are also the only population that both expand and change location, becoming intermingled with/ventral to the dI3 population. Interestingly, we see this shift in location occurring between the 25 ng/μl and 50 ng/μl condition in the *Bmp4* concentration series (data not shown). Thus, for 5 ng/μl and 25 ng/μl conditions, the dI2 population is larger following *Bmp4* electroporation, but in the correct spatial order with respect to the dI1 and dI3s, i.e. midway between them. However, by the 50 ng/μl condition (which has >2 fold more dI2s than the 25 ng/μl, *Figure 4J*) the dI2s are now ventral to dI3s. It is possible that as the dP2 Ngn1$^+$ domain is depleted, the more ventral Ngn1$^+$ domain, that would normally give rise to ventral INs (*Perez et al., 1999*), is co-opted to make dI2s. These findings suggest that the location of progenitors along the dorsal-ventral axis determines their response to BMP4: thus, the dorsal-most progenitors can receive both patterning and differentiation cues from BMP4, whereas BMP4 can only direct Ngn1$^+$ progenitors to a dI2 fate. This result in turn suggests that HH stage 15 dorsal progenitors are not equivalently competent; rather they have been positionally subdivided by an earlier patterning event, perhaps mediated by Shh or retanoids.

## The BMPs do not function as morphogens in the spinal cord

The defining characteristic of a morphogen is that it patterns cellular identity in a concentration-dependent manner. The BMPs have been widely assumed to pattern the dorsal spinal cord as a morphogen gradient by analogy with Shh signaling in the ventral spinal cord. Studies in *Drosophila* (*Kwan et al., 2016*; *Bier and De Robertis, 2015*), zebrafish (*Tribulo et al., 2003*; *Tucker et al., 2008*) and the developing telencephalon (*Watanabe et al., 2016*) have provided support that BMPs can act as morphogens. However, we found no evidence for this model in the spinal cord, either in our in vivo or in vitro studies. The spatial morphogen model predicts that the highest concentration of BMPs should specify dI1s, while a lower concentration specifies the dI2s or dI3s. We rather found that increasing the concentration of BMPs increased the efficiency by which they directed their range of cellular identities, rather than changing their identity.

The temporal morphogen model predicts that prolonged exposure to BMPs dorsalizes neural fates over time. We found no evidence for this model in our in vitro studies with mouse cells (*Figure 4—figure supplement 1*). The situation in vivo is more nuanced. A previous study found that progenitors in HH stage 10 neural fold explants were progressively dorsalized when cultured with BMP4 (19). However, the competence of progenitors to respond to the BMPs appears to have changed by HH stage 15 (~12 hr later), our electroporation time point, and the period when BMP expression begins in the chicken spinal cord (*Figure 1U–B'*). By this later stage, dorsal and ventral-dorsal progenitors show differential competence in their ability to respond to BMP4. We find that while BMP4 can have prolonged, reiterative effects on progenitors, first patterning dorsal progenitors as dP1s, and then driving dI1 differentiation. However, underlying positional information in the dorsal-most spinal cord has already directed progenitors towards being competent to assume a dI1 or dI2 fate in response to BMP4.

How widely will the BMP-signal-specific model be applicable? Many other organ systems require multiple BMPs for their development. For example, BMP2, BMP3, BMP4, BMP5 and BMP7 are expressed in the developing kidney (*Godin et al., 1999*). When two of these BMPs, BMP2 and BMP7, were assayed for their ability to induce renal branching, only BMP7 was sufficient to do so (*Piscione et al., 1997*). Similarly, many BMPs, including BMP2, BMP4 and BMP7, are expressed in the developing tooth (*Aberg et al., 1997*) and eye (*Huang et al., 2015*). In the skeletal system, only GDF6 (BMP13) and GDF7, and not BMP2, have the ability to induce tendon differentiation (*Berasi et al., 2011*). Thus, the presence of multiple BMPs in overlapping and/or adjacent locations is a general feature of organogenesis, making it feasible that BMPs will have signal-specific activities directing cell fate in other systems.

## Different BMP signaling components mediate the ability of the BMPs to promote patterning or differentiation

Our studies suggest a model where BMPs mediate their signal-specific activities by activating distinct type I Bmprs in progenitors, to permit differential progression through the cell cycle. This model concurs with previous studies demonstrating that BmprIa and BmprIb translate distinct BMP activities during development. Biochemical studies have previously shown that different BMP ligands can bind to the type I Bmprs with different affinities (*ten Dijke et al., 1994*) perhaps driving distinct biological activities. In the early embryo, activation of BmprIa can promote progenitor proliferation while BmprIb drives differentiation and apoptosis (*Panchision et al., 2001*). Both BmprIa and BmprIb direct cell fate specification in the spinal cord (*Timmer et al., 2002*), but only BmprIb is required to translate the axon guidance activities of the BMPs (*Yamauchi et al., 2008*).

This study refines our understanding of how the type I Bmprs direct dorsal spinal cell fate, by showing that BmprIa can translate the patterning activities of one set of BMPs (BMP5, BMP6, BMP7), while BmprIb promotes differentiation after activation by BMP4. The mechanisms by which activation of distinct BMP receptor complexes promote distinct effects are not known. The role of the Smad complex, the canonical second messenger for BMP signaling (*Hegarty et al., 2013*), remains unresolved. Our previous studies have demonstrated that Smad5 is the key Smad protein required for specification of cell fate (*Hazen et al., 2012*). Here, we show that each BMP can activate Smad1/5/8 to different levels, with BMP4 and BMP5 most effectively increasing pSmad1/5/8 (*Figure 2E* and *Figure 2—figure supplement 1B*). However, a specific level of pSmad5 does not appear to encode a specific cell fate, as predicted from the graded distribution of pSmad1/5/8 in

the tissue immediately adjacent to the RP (*Hazen et al., 2012*) (*Figure 2D*). Thus, while the electro-poration of both *Bmp7* and [low] *Bmp4* results in similarly low levels of pSmad1/5/8 (*Figures 2F, V, 4B and J*), these manipulations do not result in similar patterning activities and fate outcomes. Future studies will evaluate whether the type I Bmprs differentially regulate the ability of Smad5 to turn on the specific genes involved in neurogenesis and/or cell cycle control.

Our studies support the model that an output of differential BMP signaling is the regulation of the cell cycle. There is precedence for this model: previous studies have shown that GDF11 (BMP11) can inhibit progenitor proliferation in the olfactory epithelium, (*Wu et al., 2003*; *Wu and Hill, 2009*) while it promotes cell cycle exit in the developing spinal cord (*Shi and Liu, 2011*). Thus, GDF11 regulates the cell cycle in a manner consistent with promoting differentiation. Similarly, BMP4, but not BMP7, can promote neurogenesis in the olfactory epithelium (*Shou et al., 2000*). We find that several BMPs, (BMP5, BMP7 and GDF7), have shared effects on progression through the cell cycle in the chicken, that are distinct from BMP4. Similarly, BMP4 and BMP6 administration to mESC cultures exerts distinct changes in the cell cycle, while other BMPs (BMP5, BMP7 and GDF7) have no effect. These experiments are complicated to interpret, but they are nonetheless consistent with a model in which BMP4 acts as a differentiation factor, prolonging the cell cycle, distinct from action of the patterning BMPs, BMP7 and BMP6 (*Figures 6O, 7U and V*). Further experiments are needed to assess whether this mechanism is causal, for example whether these alterations in the cell cycle direct patterning, or are themselves a consequence of a patterning activity.

### BMPs can direct stem cells towards dorsal spinal identities

Finally, these studies have implications for the development of stem-cell replacement therapies to repair the spinal cord. The spinal cord is particularly vulnerable to injury, given the lack of functional redundancy with other structures. To date, significant progress has been made deriving ventral motor neurons from stem cells to permit the recovery of coordinated movement (*Wichterle et al., 2002*; *Adams et al., 2015*). However, relatively limited progress has been made generating the spinal sensory INs that integrate somatosensory information from the periphery and relay it to the brain. These studies advance that goal, by first, identifying the code by which the BMPs specify the dorsal-most sensory INs during development and then developing protocols to direct ESCs towards dorsal spinal fates. We have identified that BMP6 most effectively directs mESCs towards the RP, whereas BMP4 directs the dI1 fate, and a mixture of BMPs most effectively directs the dI3 fate (*Figure 3*, *Figure 3—figure supplement 2*). Future studies will determine whether BMP4, which is not expressed in the developing mammalian spinal cord, is nonetheless capable of generating stem cell derived dI1s that functionally mirror their endogenous counterparts. We will also continue to assess conditions to derive the dI2s. The identification of protocols to derive dI1s and dI3s in vitro have the potential to restore proprioception and pre-motor control (*Lai et al., 2016*).

## Materials and methods

### In ovo electroporation of chicken embryos

For effective expression in chicken embryos, mouse *Bmp4*, *Bmp5*, *Bmp7* and *Gdf7* inserts (*Butler and Dodd, 2003*) were subcloned in front of the CAG enhancer in the CAGGS-IRES-*Gfp* (pCIG) vector (*Megason and McMahon, 2002*). The CAG enhancer is comprised of a CMV enhancer and chicken β-actin promoter (*Miyazaki et al., 1989*). The Bmpr complex was inhibited by expressing dominant negative (dn) type I Bmprs under the control of the chicken β-actin promoter (*Lim et al., 2005*). In these constructs, the intracellular domain of the type I Bmpr was replaced with GFP.

Fertile White Leghorn eggs (McIntyre Poultry and Fertile Eggs, Lakeside, CA) were incubated for 60 hr until the embryos developed to HH stages 14–15. The spinal cord was electroporated (*Yamauchi et al., 2008*) and then allowed to develop for 48 hr until HH stages 24–26. The following constructs were used: pCIG vector (1 µg/µl), CAG::*Bmp4*-IRES-*Gfp* (5 ng/µl, 25 ng/µl, 50 ng/µl or 500 ng/µl), CAG::*Bmp7*-IRES-*Gfp* (500 ng/µl), CAG::*Gdf7*-IRES-*Gfp* (500 ng/µl) and CAG::*Bmp5*-IRES-*Gfp* (500 ng/µl), β-actin:: *dnBmprIa-Gfp* (1 µg/µl) and β-actin::*dnBmprIb-Gfp* (1 µg/µl). Information about the expression plasmid concentration for each experiment can be found in *Supplementary file 5*. In all cases, the presence of GFP demonstrates the electroporation efficiency.

To alter the concentration of *Bmp* expression: the CAG::*Bmp4* expression vector was diluted with the pCIG vector, to hold the DNA concentration constant at 500 ng/µl across experiments. *Bmp* electroporation does not generally induce the expression of other *Bmps*. Rather, only the expression of the electroporated *Bmp* is upregulated (*Figure 2—figure supplement 2*). For the cell cycle experiments, a pulse of BrdU (5'-bromo-2'deoxyuridine) was added to chicken embryos 30 min prior to dissection (*Warren et al., 2009*).

Note that the electroporation efficiency did vary slightly across experiments, resulting in some variation in the increases of cell numbers observed between experiments. However, the trend observed for the cell fate changes and the level of activation of pSmad1/5/8, the downstream effector of BMP signaling, is consistent for each BMP regardless of electroporation efficiency (*Figure 2—figure supplement 3*). High BMP concentration (>500 ng/µl), coupled with efficient electroporation could result in severe morphological changes, including the widening of the ventricle, thinning and elongating of the electroporated side of the spinal cord and the formation of a large tumorous ball of Lhx1/5$^+$ cells in the ventral spinal cord (see *Figure 2—figure supplement 4* for examples).

Quantification:

The non-electroporated side of the spinal cord could not be used as the negative control in cell-counting experiments, because *Bmp* misexpression affected patterning on both sides of the spinal cord, most notably for the dP1 and dI1 populations (compare *Figure 5H–J*). Thus, the cell number on the electroporated side was normalized against a *Gfp* control electroporation performed at the same time. Similarly, in the cell cycle experiments, the number of BrdU$^+$ cells was normalized to the area of the progenitor domain.

For the quantification of pSmad1/5/8 and Sox2 intensity, control and experimental embryos were stained on separate slides, requiring the mean intensity of the electroporated side to be normalized to the non-electroporated side of the same spinal cord after subtracting the background. Thus, these fold increases may be under-representative. Both cell numbers and staining intensity was quantified using the ImageJ software. Biological replicates: 1–2 chicken embryos per experimental condition were collected within an experiment. Each electroporation experiment was repeated at least three times. Technical replicates:>10 or more sections per embryo were analyzed. All statistical analyses were performed using either a two-tailed Student's *t*-test, if the data fit a normal distribution (Anderson-Darling test), or a Mann-Whitney test. We used the Fisher exact test on a 6 × 2 table (# of RP cells, dI1s, dI2s, dI3s, dI4s, dI5s) to assess whether *Bmp4* and *Bmp7* misexpression result in statistically similar range of cellular activities,

## Immunohistochemistry

Chicken embryonic spinal cords and mouse embryoid bodies (EBs) were thin-sectioned to yield 20 µm and 12 µm sections respectively. Antibodies against proteins were used for immunostaining and can be found in *Supplementary file 1*. Species appropriate Cyanine 3, 5 and Fluorescein conjugated secondary antibodies were used (Jackson ImmunoResearch Laboratories). Images were collected on Carl Zeiss LSM700 and LSM800 confocal microscopes.

## In Situ hybridization

In situ hybridizations were performed on chicken (HH stage 14–26) and mouse (E9.5–10.5) embryonic spinal cords. 3'UTR probes were designed using http://primer3plus.com and verified for specificity to the gene of interest using http://www.ncbi.nlm.nih.gov/tools/primer-blast/.

Chicken and mouse primer sequences used to make in situ hybridization probes can be found in *Supplementary file 2* and *3* respectively. Probes were made using a DIG RNA labeling kit (Roche, Indiananapolis, Indiana). Images were collected on a Carl Zeiss AxioImager M2 fluorescence microscope with an Apotome attachment.

## Mouse embryonic stem cell culture

MM13 mESCs (a gift from T. Jessell, Columbia University (*Wichterle et al., 2002*) which test negative for mycoplasma contamination) were grown on irradiated mouse embryonic fibroblasts and maintained in a proliferative state using Leukemia inhibitory factor in an embryonic stem cell media. mESCs were differentiated in suspension to form EBs (*Wichterle et al., 2002*). 1 µM retinoic acid (RA) was added to the media at day 1 of differentiation to direct cells toward a spinal neural identity

(*Muto et al., 1991*). 24 hr later, BMP recombinant proteins (R and D Systems; Invitrogen) were added to the media to direct EBs toward a dorsal identity. Information about the concentration of recombinant proteins used in each experiment can be found in *Supplementary file 5*. EBs were collected at day nine for the concentration-comparison experiments, or every 3 days from day 0 to day 15 for the time-course experiments. EBs were either fixed with 4% PFA and processed for immuno-histochemistry or RNA was extracted (Qiagen RNeasy kit) for qRT-PCR. For BrdU incorporation experiments BrdU was added to the media 2 hr prior to collection at day 9, then fixed with 4% PFA and processed for immunohistochemistry as before.

Dorsomorphin and LDN-193189 (Stemgent) were used to inhibit BMP signaling. Low concentrations of Dorsomorphin (1 uM) have been shown to block both the activin type IA receptor (Acvr1 or Alk2) and BmprIa (Alk3), while higher concentrations of Dorsomorphin (10 uM) additionally block BmprIb (Alk6) (*Yu et al., 2008b*). Similarly low concentrations of LDN-193189 (100 nM) block only Acvr1 and BmprIa, while higher concentrations (500 nM) also block BmprIb, activin type IB receptor (Acvr1B or Alk4), TGFβ receptor 1 (Tgfbr1 or Alk5) and activin type IB receptor (Acvr1C or Alk7) (*Yu et al., 2008a*).

We assessed whether either the timing of RA and BMP addition, or manipulation of the Wnt and Shh signaling pathways, improved our ability to produce more dorsally directed EBs. However, modifying the timing of RA and BMP addition had no effect on axial level or progenitor/IN identity (*Figure 3—figure supplement 1A–B*). Similarly, neither altering Wnt signaling nor inhibiting Shh signaling had an effect on the efficiency of dorsal IN differentiation (*Figure 3—figure supplement 1A*).

## qRT-PCR experiments

RNA collected from EBs was reverse transcribed to make cDNA (Superscript III kit, Invitrogen). cDNA was analyzed for gene expression using quantitative (q) PCR. qPCR was performed on a Roche 480 Lightcycler using Roche enzymatic reagents (SYBR green I Master mix). Primers were designed using the qPCR setting on http://primer3plus.com and verified for specificity using http://www.ncbi.nlm.nih.gov/tools/primer-blast/. Relative fold change in expression was calculated using the 2(-ΔΔC(T)) method (*Livak and Schmittgen, 2001*) comparing the gene of interest to the control housekeeping gene, Glyceraldehyde 3-phosphate dehydrogenase (GapDH). Biological replicates: dozens of EBs within a treatment group were collected for RNA extraction during each experiment; at least 2 independently cultured wells of EBs were collected to make 2 samples of RNA per group. Complete mESC differentiation experiment were repeated two or more times. Technical replicates: at least two cDNA reactions were performed per RNA sample; cDNA samples were run in triplicate and qPCR was performed on each cDNA sample at least twice. All statistical analyses were performed using a two-tailed Student's *t*-test. Primer sequences can be found in *Supplementary file 4*.

## Acknowledgements

We are most grateful to Jeff Golden, Thomas Jessell, Ed Laufer, Thomas Müller, Bennett Novitch for reagents and Daniel Sivalingam for early technical help. We would also like to thank Tanmoy Bhatta-charya, Karen Lyons, Harley Kornblum, Bennett Novitch and Sandeep Gupta, as well as members of the Butler and Novitch laboratories past and present, for invaluable discussions and comments on the manuscript. This work was supported by fellowships from the Broad Center for Regenerative Medicine at UCLA and the Rose Hills Foundation and the NICHD T32 training grant (HD060549) in Developmental Biology, Stem Cells and Regeneration to MGA, fellowships from the California Institute for Regenerative Medicine (CIRM) Bridges to Research program (TB1-01183) for LdC and EO-B and grants from CIRM (RB5-07320), the National Institute of Health (NIH) (NS085097) and the UCLA Broad Stem Cell Research Center to SJB.

## Additional information

### Funding

| Funder | Grant reference number | Author |
| --- | --- | --- |
| National Institutes of Health | R01: NS085097 | Samantha J Butler |
| California Institute for Regenerative Medicine | RB5-07320 | Samantha J Butler |
| UCLA Broad Stem Cell Research Center | Research Grant | Samantha J Butler |
| National Institutes of Health | T32 training fellowship: HD060549 | Madeline G Andrews |
| California Institute for Regenerative Medicine | Bridges to Research fellowship: TB1-01183 | Lorenzo M del Castillo Eliana Ochoa-Bolton |

The funders had no role in study design, data collection and interpretation, or the decision to submit the work for publication.

### Author contributions

Madeline G Andrews, Conceptualization, Data curation, Formal analysis, Validation, Investigation, Methodology, Writing—original draft, Project administration, Writing—review and editing; Lorenzo M del Castillo, Formal analysis, Investigation, Methodology; Eliana Ochoa-Bolton, Ken Yamauchi, Jan Smogorzewski, Investigation, Methodology; Samantha J Butler, Conceptualization, Data curation, Formal analysis, Supervision, Funding acquisition, Methodology, Writing—original draft, Project administration, Writing—review and editing

### Author ORCIDs

Madeline G Andrews  http://orcid.org/0000-0002-5154-5081
Samantha J Butler  http://orcid.org/0000-0002-9491-7551

### Decision letter and Author response

Decision letter https://doi.org/10.7554/eLife.30647.032
Author response https://doi.org/10.7554/eLife.30647.033

## Additional files

### Supplementary files

• Supplementary file 1. Antibody information
DOI: https://doi.org/10.7554/eLife.30647.026

• Supplementary file 2. Chicken primer sequences for in situ hybridization experiments
DOI: https://doi.org/10.7554/eLife.30647.027

• Supplementary file 3. Mouse primer sequences for in situ hybridization experiments
DOI: https://doi.org/10.7554/eLife.30647.028

• Supplementary file 4. Mouse primer sequences for qRT-PCR
DOI: https://doi.org/10.7554/eLife.30647.029

• Supplementary file 5. BMP concentrations used in these studies
DOI: https://doi.org/10.7554/eLife.30647.030

• Transparent reporting form
DOI: https://doi.org/10.7554/eLife.30647.031

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
