## [Decision Letter]

[Editors’ note: a previous version of this study was rejected after peer review, but the authors submitted for reconsideration. The first decision letter after peer review is shown below.]

Thank you for submitting your work entitled "BMPs direct sensory interneuron identity in the developing spinal cord using signal-specific not morphogenic activities" for consideration by *eLife*. Your article has been reviewed by three peer reviewers, and the evaluation has been overseen by Marianne Bronner as Senior Editor. The reviewers have elected to remain anonymous.

Our decision has been reached after extensive consultation between the reviewers. Based on these discussions and the individual reviews, we regret to inform you that your work will not be considered further for publication in *eLife*. You will see that all three reviewers see merit in the work but also raise considerable concerns that will require a good deal of additional work to address.

Specifically, the reviewers agreed that it is important to address how BMPs are directing neuronal diversity. However, they all felt that it would take substantial new experiments with different concentrations of BMPs in the chick system, as well as providing evidence for the different concentrations, to satisfy their concerns. Also, the effects on cell cycle need to be more convincing. We hope that you find the detailed comments of the reviewers below helpful in revising the paper.

Reviewer #1:

Here the authors address the question as to how BMPs operate to generate specific populations of dorsal interneurons in the developing spinal cord. Current models suggest that BMPs act as classical morphogens that direct progenitor patterning and subsequent neuronal fates through graded signals; however definitive evidence for this model is lacking. Here the authors aim to distinguish whether BMPs are indeed morphogens in this capacity or conversely, have defined and specific functions in directing progenitor patterning and neuronal differentiation. This is an interesting question and would provide new insight into the function of BMPs in patterning.

The experiments that directly test the central hypothesis utilize two systems, in ovo electroporation in chick spinal cords and differentiated mouse embryonic stem cells. The strength of the in vitro cultures is that the concentration of BMPs can be carefully titrated and effects on dorsal interneuron (dI) generation can be evaluated (much like the original experiments testing effects of Shh on ventral neuron specification). The result that distinct BMPs have specific roles in dI generation that appears independent of concentration is convincing in the culture system (Figure 3 and Figure 4). Unfortunately, the chick experiments are problematic. First, the chick experiments are performed in the context of gradients of endogenous BMP proteins, complicating interpretation of the effects of exogenous BMP overexpression (how exogenous BMPs affect endogenous gradients and signal transduction is not discussed and not determined here. Authors state in the Materials and methods that BMP electroporation does not generally induce expression of other BMPs, but does this apply to all BMPs? What happens to gradients?). Insets shown in Figure 2 show amount of transfection read out by GFP (I assume), and these images show quite a lot of variation in transfection efficiency. Given that the amount of transfection will dictate levels of ligand, how do the authors control for transfection efficiency and rule out the possibility that outcomes are a consequence of differential levels of expression?

The key experiment to manipulate levels of BMPs through electroporation consists of electroporating two different amounts of expression plasmid (Figure 4). It is not clear how the authors controlled for transfection efficiency, which would directly affect the amount of BMPs generated in the embryo. The amounts of plasmid transfected are also not clear: the text reads that the lower BMP condition consists of a "1:9 mixture of CAG::*gfp* and CAG::*BMP* vectors", which implies 90% of the "high" concentration. However, the Materials and methods section suggests the reverse i.e. 9:1, which would be 10% of the high concentration. Nevertheless, in the best case scenario, the authors are addressing two concentrations of BMP termed "high" and "low" (40% and 15% of pSMAD signal output) and both lead to the same effect but with different efficiencies. The authors interpret this to mean that BMPs do not act as a morphogen in this context. An alternative interpretation is that these two concentrations have the same operational output and that going lower might have different outcomes. It seems important for the authors to test a range of amounts before arriving at this conclusion. I agree that the mouse cultures support their interpretation but, different BMPs are utilized in the mouse compared with chick, and as such mouse BMPs could have different modes of function during patterning and differentiation.

A second issue in this study is the study of cell-cycle dynamics. The authors state that a patterning factor might speed up cell-cycle transitions, but this does not have to be the case. The authors measure BrdU incorporation in specific progenitor domains after electroporation of BMPs. They argue that BMP4 and BMP7 have differential effects on proliferation according to different domains. However, the authors show that treatment with BMPs can have dramatic consequences on expanding or almost eradicating these same progenitor domains (outlines in Figure 6 show this clearly). Clearly, these expansions and reductions need to be taken into account to avoid skewing the BrdU counts. However, it is not clear how the counts were done. This is important to know. The authors make some strong statements about BMPs regulating cell-cycle length and cell-cycle exit, and use these conclusions as basis for their ideas on mechanism. However, supporting evidence is weak. While changes in BrdU^+^ cells or pH3+ cells are suggestive of changes in cell-cycle length/exit, it is important to measure both directly. This is difficult to do in the chick system (due to the closed environs of the egg-but perhaps can be addressed in NEW cultures) but can be done in mice. This is likely to be outside the scope of the current study, but it forms an essential part of the authors' argument. Without this, the models drawn in Figure 7 are an overstatement (also, did not see evidence in this study that BMPr1a is involved in roof plate formation – Figure 7).

Finally, it is hard to draw parallels between the chick studies and the differentiated EBs in culture in order to gain insight into the question of how BMPs function (morphogen or role-specific) because of involvement of distinct BMPs in the different species. The chick experiments point to overlapping/functions of BMP4 and BMP7 in many instances, so it is hard to make the point that they have specific roles in patterning and dI development. The culture system seems to be a better starting point to address this question.

Reviewer #2:

This study challenges the dogma that BMP gradients/concentrations and/or temporal characteristics of exposure specify the dorsal neural tube neuronal populations, particularly dI1-dI3. Using the chick neural tube and comparisons with mouse ES cells, they show distinct functions for BMP4, 5, 6, 7 and GDF7, particularly highlighting BMP4 and BMP7. These BMPs have different effects on the cell cycle of progenitors, the expression of bHLH factors that define the progenitors, and the dI populations that arise from these progenitors. Differences in use of BMPs in mouse and chick are identified. For the most part, the results support the conclusions. I have suggestions on clearing up some inconsistencies in presentation. Also, the results showing the effect of the BMPs on the cell cycle are not strong so conclusions should be tempered or better experiments performed.

1) I have a problem with Figure 4 where 2 concentrations of expression plasmid for BMP4 are being tested in the chick electroporation assay. Here, rather than cell number (as is the norm and is used in Figure 2), they quantify the data by fluorescence intensity. This is particularly a problem for the dI2 population that is identified by expression of Lhx1/5 but an absence of Pax2. Is the intensity from the Pax2+ cells subtracted out from the Lhx1/5 signal? This is not clear. Furthermore, from the image (Figure 4), it looks like Pax2 is suppressed (loss of dI4), thus leading to more cells defined as dI2-but are they really? This needs clarification for why change the quantification (cell number is best here I believe), and how to interpret the loss of dI4.

2) Measuring the length of the progenitor domain based on the in situs for Atoh1, Ngn1 and Ascl1 are questionable. It may be fine for Atoh1 (clear increase with BMPs), and Ngn1 (striking loss with BMP4), but the Ascl1 is not convincing. If anything it looks like the domain is shifted down, not necessarily expanded. This is also pretty crude given the morphological disruption of the neural tube seen. More caution in concluding anything from this is suggested. This also affects the BrdU experiment where the in situ is used to highlight the particular progenitor domain. This is not very accurate. Really need double immuno for the progenitor markers and BrdU, which is technically challenging. With this in mind, the cell cycle conclusions are not compelling.

3) No effort to show what levels of BMP4 or BMP7 are attained in the chick electroporation assays. Authors need to show if they are expressed at comparable levels when the same amount of plasmid is used given conclusions are being made about specificity of activity not concentration being important. For the in vitro data this is not a question since the BMPs themselves are added to the media at defined concentrations.

4) Some discussion of what has been done in the BMP field to match ligands with receptors-has anything been done? This concept that the different BMPs have specificity for receptors is reported. More reference to what has been done in the field along these lines seem relevant. Also, there is a whole literature on BMP gradients specifying cell type in zebrafish. Is there just a difference is species or has it not been addressed in experiments in fish. Work from Mary Mullins comes to mind.

Reviewer #3:

The main crux of this study is that different BMPs act differentially to direct dorsal progenitor and interneuron identity and do not act as a set of classical morphogens, working through concentration and time to progenitor lineage restriction and specify cell fate.

The data seem reasonable for a differential effect on interneuron identity but I wasn't so convinced by the effects on progenitor identity. I also wonder whether a slightly different statistical analysis might not tease things out better and make the discussion easier- namely something like the Kolmagorov-Smirnov test of differences in distribution. My understanding is that different BMPs should have differing distributions of effects on the different dorsal interneuron populations and so an analysis of these distribution- rather than a direct comparison of one cell type by different BMPs might be a better way to go forward.

I go through each figure in turn with comments.

Figure 1. This seems completely reasonable in terms of an analysis of gene expression at different stages in the chick.

Figure 2 was slightly puzzled by the upregulation of phospho smad on the contralateral hemi-spinal cord with BMP4 electroporation in E.

I wasn't sure of how the authors distinguished dI4 from dI6 in their summary between N and O- indeed there is no mention of dI6 in the quantitation shown in V. Could it be possible that there is a dramatic reduction in dI6 with BMP4 or a dramatic increase in dI4 with BMP7? The results for dI3 and 5 look convincing. I did notice that the spatial organisation of these distinct interneurons also seemed perturbed- so not just number but position could also be altered by changing BMP expression.

In V- I wonder if an analysis of the distribution of interneuron identities following the different perturbations might be informative- i.e. is the distribution of RP and dorsal interneuron types changed differentially by BMP4 and BMP7 or are both distributions similar?

Figure 3 shows quantitation of RT q-PCR results on embryoid bodies treated with different BMPs suggesting that BMP4 upregulates dI1 markers preferentially, in comparison to the other BMP family members. This data looks convincing and the distributions (see above) look clearly different. I do wonder whether RT-1-PCR is the best measure of this though- rather than the protein expression (cellular) analysis shown in other parts of the paper.

Figure 4. Here- changes to concentration of plasmids electroporated are used as a proxy for changes in BMP concentration. This assays, quite rightly, by phospho SMAD expression. It would be nice to see some kind of quantitation of the effects of the different plasmid concentrations on SMAD expression. Again, the distributions of dorsal interneuron identities generated by these different perturbations do look different but it might be nice to do this direct comparison through the same concentration of plasmid across the different interneuron types to see if the different concentrations display the same distributions.

Figure 5. Here- I wasn't really sure on the differences in progenitor characteristic and identity that were being displayed. The progenitor domains seemed the same, in terms of their distributions following BMP4 and BMP7 misexpression but I might be missing something subtle here.

Figure 6. I wonder if different pulse-lengths of BrdU might be informative to look at changes in cell cycle following different BMP exposures.

Finally, in Figure 7 – I wasn't clear on the expression of the BMPR1b in the dorsal spinal cord early in development. Is it expressed in the right time and right place to mediate differential signalling dependent on the BMP being exposed in the wild-type.

[Editors’ note: what now follows is the decision letter after the authors submitted for further consideration.]

Thank you for resubmitting your work entitled "BMPs direct sensory interneuron identity in the developing spinal cord using signal-specific not morphogenic activities" for further consideration at *eLife*. Your revised article has been favorably evaluated by Marianne Bronner as Senior and Reviewing editor, and three reviewers.

The manuscript has been improved but there are some remaining issues that need to be addressed before acceptance, as outlined below. Although two of the reviewers are completely satisfied, reviewer 3 requested more experiments. I've included the full review below for your information. However, after discussion between the reviewers, it was agreed that textual changes would be sufficient to address their concerns.

Reviewer 3 was not satisfied with the claims regarding cell cycle exit and would like the following changes:

The authors should resist from making firm conclusions about the length of the cell-cycle and the ability to promote exit from the cell-cycle, and should tone down the text regarding these points. They should also remove Figure 6 and Figure 7 as these models are misleading. The models in Figure 8 is a problem, but as long as the text tones down the cell-cycle effects and states that other experiments are needed when discussing the final over-arching model, it will be OK.

Full comments of reviewer #3:

The authors have provided a revised version of their manuscript including new data that address concerns regarding the chick electroporation experiments. Specifically, they include tests of a broader concentration of plasmids expressing BMP4, and focus more on cell-cycle experiments in the mouse in vitro system. The additional experiments for the mouse system complement the mouse analyses and lend further support to the investigators findings. As mentioned in the earlier review, the observations that the investigators perform in the mouse system are solid and support their hypothesis that BMPs in this paradigm, do not function as simple morphogens. However, the experiments in the chick still remain a problem. The investigators confirm in the text and in their response letter that the chick and mouse system are indeed different (see response point 2) and that there are caveats to the in vivo experiments. I am not sure how expanding the mouse experiments helps assuage the problems with the chick. Overall, while the manuscript is improved, the core issues that I had with the initial submission have not been fully addressed, and there are sufficient discrepancies within the revised version that dampen enthusiasm.

Some problems that I still have with the chick experiments are detailed below:

1) The authors include a nice array of plasmid concentrations to evaluate the effects of BMP4; but the question about the effects on endogenous BMPs remain. The authors show that BMP transcription is not changed, but there is no demonstration that endogenous BMP protein gradients are not altered upon BMP overexpression.

2) The authors talk about fate changes in response to BMPs- I am wary of the use of this term in this context because the authors show that BMPs affect cell-cycle. The observations of increased dI numbers in response to BMPs could be explained by proliferative changes within individual domains. I agree that there is an expansion of the di1 and dI3 domains, but this could also be due to proliferation and not fate. The reason that I am wary of the use of the word "fate" or "specify" is that if there were indeed a fate change, then one would expect reductions in neighboring populations but the amounts of the neighboring IN populations are either increased or remain largely the same. I would welcome some clarification of this point.

3) The cell-cycle studies are still very problematic. I still think that the chick experiments are over-interpreted (see the model in Figure 6) without more detailed experiments. In the chick, BMP4 expression causes no change in S-phase cell numbers and increased numbers of cells in M-phase-the authors interpret this as increased number of dividing cells with increased propensity for cell-cycle exit. I am not sure why they bring in the cell-cycle exit affect: there is no change in progenitor:neuron ratio as stated by the authors, and morphological changes elicited are consistent with increased proliferation. I have reservations about the model as it suggests that BMP4 promotes neural differentiation, when there is no evidence for this. Could all be explained by increased proliferation and normal exit of cell-cycle.

Further, expression of BMP7 causes reduced numbers of cells in S-phase and increased M phase- there is no real basis here to conclude that the cell-cycle is shortened simply by the increased numbers of cells in M-phase; it is a good first observation but really warrants further analysis to obtain a firm conclusion.

The authors approach for deeper validation of the chick studies is to repeat the cell-cycle experiments in the mouse system, but as mentioned previously in the original review, this is a vastly different system, and different BMPs are in play. Speaking to this point, the results in the mouse system appear different from the chick experiments. In the mouse system, BMP4 causes a marked increase in the numbers of S phase cells (not seen in chick) and also M-phase cells-suggesting increased proliferation, and not supporting the increased propensity for cell-cycle exit suggested by the authors for the chick. BMP6 (they hypothesize that this is equivalent to BMP7 in chick) in this system now only increases cells in S-phase (not a reduction as they see in chick) and no change in M-phase cells (they see an increase in the chick). Some possibilities to consider are 1. BMP6 in the mouse is not the BMP7 homologue as the authors put forward; 2. It is the BMP7 homologue and the effects are simply different.

The authors state in the text that their in vitro studies are consistent with the in vivo studies. While I agree that BMP addition in the mouse system does indeed alter cell-cycle dynamics, there are significant differences that could have different functional outcomes. These differences underscore the difficulty in extrapolating from the chick to the mouse in vitro system.

---

## [Author Response]

[Editors’ note: the author responses to the first round of peer review follow.]

Specifically, the reviewers agreed that it is important to address how BMPs are directing neuronal diversity. However, they all felt that it would take substantial new experiments with different concentrations of BMPs in the chick system, as well as providing evidence for the different concentrations, to satisfy their concerns. Also, the effects on cell cycle need to be more convincing. We hope that you find the detailed comments of the reviewers below helpful in revising the paper.

We did indeed find the detailed comments from the reviewers below extremely helpful in revising the manuscript. In brief, the most substantial amendments to the manuscript include:

Figure 4 more complete series of BMP concentrations which demonstrates more convincingly that while higher [BMP4] are more efficient in directing cell fate, all [*Bmp4*] direct the same range of cellular effects.

Figure 7 (new): New mESC studies, which demonstrate that the BMP4 and BMP6 also have distinct effects on the cell cycle in vitro.

Figure 8 (previously Figure 7): addition of HH stage 21 ISH and quantitation

Figure 2—figure supplement 2: to show that BMP misexpression is specific, it alters the expression of only the electroporated BMP

Figure 2—figure supplement 3: to demonstrate that BMP misexpression results in consistent levels of pSmad1/5/8

Supplementary file 5 detailed description of all the BMP concentrations tested in the studies.

Reviewer #1:

Here the authors address the question as to how BMPs operate to generate specific populations of dorsal interneurons in the developing spinal cord. Current models suggest that BMPs act as classical morphogens that direct progenitor patterning and subsequent neuronal fates through graded signals; however definitive evidence for this model is lacking. Here the authors aim to distinguish whether BMPs are indeed morphogens in this capacity or conversely, have defined and specific functions in directing progenitor patterning and neuronal differentiation. This is an interesting question and would provide new insight into the function of BMPs in patterning.

The experiments that directly test the central hypothesis utilize two systems, in ovo electroporation in chick spinal cords and differentiated mouse embryonic stem cells. The strength of the in vitro cultures is that the concentration of BMPs can be carefully titrated and effects on dorsal interneuron (dI) generation can be evaluated (much like the original experiments testing effects of Shh on ventral neuron specification). The result that distinct BMPs have specific roles in dI generation that appears independent of concentration is convincing in the culture system (Figure 3 and Figure 4). Unfortunately, the chick experiments are problematic. First, the chick experiments are performed in the context of gradients of endogenous BMP proteins, complicating interpretation of the effects of exogenous BMP overexpression (how exogenous BMPs affect endogenous gradients and signal transduction is not discussed and not determined here. Authors state in the Materials and methods that BMP electroporation does not generally induce expression of other BMPs, but does this apply to all BMPs? What happens to gradients?).

We agree with the reviewer that the chicken experiments are problematic for precisely the reasons they lay out, i.e. that they are performed in a background of endogenous BMP signalling. This is why we couple our in vivostudies with an in vitro approach, where it is possible to more carefully control manipulating BMP signalling. We have included a more substantive discussion of these issues in the Discussion subsection “BMPs have distinct roles directing dorsal spinal fates.

Nonetheless we believe it is important to include the in vivo approach for the following reasons:

a) The distribution of the endogenously expressed *Bmp* is not altered following in ovo electroporation of a different ectopic *Bmp*. Thus, *Bmp7* expression is not upregulated in the presence of ectopic *Bmp4* and vice versa. This result suggests that the phenotype observed we see are largely (solely?) a consequence of the ectopic *Bmp*. We now include this data in Figure 2—figure supplement 2.

b) We find that *Bm*p misexpression drives pSmad1/5/8 to a specific level throughout the spinal cord, consistent with the model that the ectopically expressed *Bmp* “overwhelms” endogenous BMP signalling. As a side note: we find it interesting that a given level of pSmad is not tuned to a specific cellular identity, rather low [BMP4] → low pSmad results in a different series of cellular fates than high [BMP7] → low pSmad.

c) There are substantial commonalities in the mechanism in the in vivo and in vitro – that BMPs direct a specific range of cell fates (in the presence or absence of endogenous signalling), they differentially affect progression through the cell cycle and that these fates are mediated by different type I Bmprs.

Insets shown in Figure 2 show amount of transfection read out by GFP (I assume), and these images show quite a lot of variation in transfection efficiency. Given that the amount of transfection will dictate levels of ligand, how do the authors control for transfection efficiency and rule out the possibility that outcomes are a consequence of differential levels of expression?

The reviewer is correct that there is some variation in the electroporation efficiency, although it is generally easiest to get high levels of electroporation in the dorsal spinal cord. Nonetheless the exact identity of the electroporated cells will differ embryo to embryo. This stochasticity can be of concern when manipulating autonomously required signalling pathways.

However, here we are examining the consequence of manipulating the levels of a diffusible ligand, i.e. it acts non-autonomously. We can see those non-autonomous effects very clearly i.e. the ability of the Bmps to influence the non-electroporated side of the spinal cord. Thus, the identity of the source cells is probably not that important, rather what is critical is the absolute level of ligand that directs the efficiency of the response. To demonstrate this more clearly we have included a Figure 2—figure supplement 3, which shows that we get similarly uniform levels of pSmad1/5/8 even with different patterns of *Bmp* electroporation.

The absolute level of the ligand will, of course, also vary embryo to embryo, however the trend across years of experiments on this project is clear and unvarying – if high levels of a given Bmp are electroporated, we get more cells adopting the same range of cellular fates. The new data in Figure 4 speaks to this point, when we increase the level of electroporated *Bmp4*, we progressively increase the level of Smad signaling and concomitantly the number of cells that adopt dI1, dI2 and dI3 fates. The range of cellular fates does not change as the ligand concentration changes as would be predicted by the canonical morphogen model.

[…] The amounts of plasmid transfected are also not clear: the text reads that the lower BMP condition consists of a "1:9 mixture of CAG::gfp and CAG::BMP vectors", which implies 90% of the "high" concentration.

We apologize for this typographical error, which has now been corrected in the text.

However, the Materials and methods section suggests the reverse i.e. 9:1, which would be 10% of the high concentration. Nevertheless, in the best case scenario, the authors are addressing two concentrations of BMP termed "high" and "low" (40% and 15% of pSMAD signal output) and both lead to the same effect but with different efficiencies. The authors interpret this to mean that BMPs do not act as a morphogen in this context. An alternative interpretation is that these two concentrations have the same operational output and that going lower might have different outcomes. It seems important for the authors to test a range of amounts before arriving at this conclusion.

We thank the reviewer for suggesting this experiment. As described in the Results section, we have now examined a wider concentration range i.e. a 99:1 ratio of CAG::*Gfp*: CAG::*Bmp4*, 19:1, 9:1 and 100% CAG::*Bmp4*.

The data is shown in Figure 4, and strongly supports our hypothesis: this concentration range resulted in a relatively linear increase in Smad1/5/8 activity and an increase in same range of cellular fates. Thus, lower levels of BMP4 did not promote a more ventral-dorsal identity as predicted by the morphogen models. Rather, concentration appears to control the efficiency by which BMP4 can direct the same range of cell fates.

I agree that the mouse cultures support their interpretation but, different BMPs are utilized in the mouse compared with chick, and as such mouse BMPs could have different modes of function during patterning and differentiation.

Reviewer 1 is correct, and indeed we think that it is likely that there are species-specific differences in the code of BMP activities that direct cell fates. However, the common themes are also important: that BMPs direct a specific ranges of cell fates, they differentially affect the cell cycle and appear to function through specific type I Bmprs to mediate specific fates.

We have added more discussion on this point in the Discussion section.

A second issue in this study is the study of cell-cycle dynamics. The authors state that a patterning factor might speed up cell-cycle transitions, but this does not have to be the case. The authors measure BrdU incorporation in specific progenitor domains after electroporation of BMPs. They argue that BMP4 and BMP7 have differential effects on proliferation according to different domains. However, the authors show that treatment with BMPs can have dramatic consequences on expanding or almost eradicating these same progenitor domains (outlines in Figure 6 show this clearly). Clearly, these expansions and reductions need to be taken into account to avoid skewing the BrdU counts. However, it is not clear how the counts were done. This is important to know.

We apologize for not clearly describing how these experiments were quantified. We normalized the BrdU^+^ cell counts to the area of the progenitor domain. This normalization was necessary as reviewer 1 correctly points out, because *Bmps* have such profound effects on the size of the progenitor domains. We have added more detailed information to the Materials and methods for readers to better assess how the quantification was performed.

The authors make some strong statements about BMPs regulating cell-cycle length and cell-cycle exit, and use these conclusions as basis for their ideas on mechanism. However, supporting evidence is weak. While changes in BrdU^+^ cells or pH3+ cells are suggestive of changes in cell-cycle length/exit, it is important to measure both directly. This is difficult to do in the chick system (due to the closed environs of the egg-but perhaps can be addressed in NEW cultures) but can be done in mice. This is likely to be outside the scope of the current study, but it forms an essential part of the authors' argument. Without this, the models drawn in Figure 7 are an overstatement.

We have added an additional analysis to these studies – new Figure 7 – examining the effect of the different BMPs on the cell cycle in our mESC cultures. The results are striking – there are ~2-fold more dividing cells (both M and S phase) in BMP4-treated cultures, while BMP6 alone of the rest of the BMPs, modestly upregulates the number of cells in S phase. Thus, BMPs have distinct effects on the cell cycle bothin vivo and in vitro. We have clarified in the text in subsection “Different BMP signaling components mediate the ability of the BMPs to promote patterning or differentiation” that this – the differential effect of the BMPs on cell cycle progression – is the most circumspect interpretation of our data.

Also, did not see evidence in this study that BMPr1a is involved in roof plate formation-Figure 7.

This experiment was performed in the mESC culture system. We showed that with low concentrations of either BMP inhibitor, i.e. when we are blocking BmprIa, there is a significant reduction in the ability of BMP6 to direct EBs to express a RP marker (Msx1).

Finally, it is hard to draw parallels between the chick studies and the differentiated EBs in culture in order to gain insight into the question of how BMPs function (morphogen or role-specific) because of involvement of distinct BMPs in the different species. The chick experiments point to overlapping/functions of BMP4 and BMP7 in many instances, so it is hard to make the point that they have specific roles in patterning and dI development. The culture system seems to be a better starting point to address this question.

As discussed in point 2 and 6 above, we have added more discussion about the interpretations of our chicken results in the Discussion section.

Reviewer #2:

This study challenges the dogma that BMP gradients/concentrations and/or temporal characteristics of exposure specify the dorsal neural tube neuronal populations, particularly dI1-dI3. Using the chick neural tube and comparisons with mouse ES cells, they show distinct functions for BMP4, 5, 6, 7 and GDF7, particularly highlighting BMP4 and BMP7. These BMPs have different effects on the cell cycle of progenitors, the expression of bHLH factors that define the progenitors, and the dI populations that arise from these progenitors. Differences in use of BMPs in mouse and chick are identified. For the most part, the results support the conclusions. I have suggestions on clearing up some inconsistencies in presentation. Also, the results showing the effect of the BMPs on the cell cycle are not strong so conclusions should be tempered or better experiments performed.

We thank the reviewer for their support. In response to their concerns, we have performed a major new experiment: we have assessed the effect of the different BMPs on the cell cycle in our mESC cultures (new Figure 7). We monitored our RA, BMP4, BMP5, BMP6, BMP7 and GDF7 treated cultures for both BrdU incorporation and pHistoneH3 staining. As discussed above, we find that there are ~2-fold more dividing cells (both S and M phase) in BMP4-treated cultures compared to RA control. In contrast, BMP6, alone of the rest of the BMPs, modestly increases the number of cells in S phase. Thus, similar to our results in vivo, BMP4 has the most profound effect on the cell cycle, with BMP6 being the other active BMP (c.f. BMP7 in chick).

As requested, we have also tempered our conclusions to clarify the key point: that the BMPs have different activities promoting cell cycle progression. How these differential effects on the cell cycle results in the specification of cell fate will require further study. One possibility will be to dissect how the activities of BMPs change over time, with respect to the cell cycle, using our in vitro mESC cultures. Do BMPs have differentiation effects as the commitment status of the cells change?

1) I have a problem with Figure 4 where 2 concentrations of expression plasmid for BMP4 are being tested in the chick electroporation assay. Here, rather than cell number (as is the norm and is used in Figure 2), they quantify the data by fluorescence intensity. This is particularly a problem for the dI2 population that is identified by expression of Lhx1/5 but an absence of Pax2. Is the intensity from the Pax2+ cells subtracted out from the Lhx1/5 signal? This is not clear. Furthermore, from the image (Figure 4), it looks like Pax2 is suppressed (loss of dI4), thus leading to more cells defined as dI2-but are they really? This needs clarification for why change the quantification (cell number is best here I believe), and how to interpret the loss of dI4.

We apologize for the misunderstanding. The previous graph in Figure 4 had two y-axes, one side was measuring cell number (which is the correct measurement as reviewer 2 points out) and the side was measuring the fluorescence intensity of pSmad. We recognize how confusing that was and have separated the graphs to make it clear to the reader what we are quantifying.

2) Measuring the length of the progenitor domain based on the in situs for Atoh1, Ngn1 and Ascl1 are questionable. It may be fine for Atoh1 (clear increase with BMPs), and Ngn1 (striking loss with BMP4), but the Ascl1 is not convincing. If anything it looks like the domain is shifted down, not necessarily expanded. This is also pretty crude given the morphological disruption of the neural tube seen. More caution in concluding anything from this is suggested. This also affects the BrdU experiment where the in situ is used to highlight the particular progenitor domain. This is not very accurate. Really need double immuno for the progenitor markers and BrdU, which is technically challenging. With this in mind, the cell cycle conclusions are not compelling.

We have now substituted the quantifications of area of these progenitor domains in Figure 5, instead of length. The result remains the same. Reviewer 2 is correct that the effect of *Bmp* misexpression on the *Ascl1* domain is smaller than with the other two domains – however it is nonetheless a significant one. The *Ascl1* domain is both slightly larger and shifted down – we have made the magnitude of the effect clearer in subsection “BMP4 and BMP7 have differential effects on dorsal progenitor identity in vivo” of the Results.

The BrdU experiment was indeed technically challenging, note that we usedin situ hybridisation experiments to define the boundary of the progenitor domain, so we could then normalize the results with respect to area.

As discussed above – we have added a further cell cycle experiment, using our in vitro approach – which show that BMP4 results in marked differences in the numbers of cells in mitosis compared to control. We have also now been more cautious discussing the cell cycle experiments.

3) No effort to show what levels of BMP4 or BMP7 are attained in the chick electroporation assays. Authors need to show if they are expressed at comparable levels when the same amount of plasmid is used given conclusions are being made about specificity of activity not concentration being important. For the in vitro data this is not a question since the BMPs themselves are added to the media at defined concentrations.

This question is surprisingly difficult to address given the stochastic nature of in ovo electroporation. We have tried using the BRE reporters of BMP activity, but found them to be inconsistent in our control experiments i.e. there is sufficient noise in the controls, that it is hard to distinguish the “real” effects of ectopic BMP expression.

However, we have no concern that high levels of BMP expression are being achieved in both cases, because of the profound – but very different! – effects that high concentrations of each expression plasmid have on the morphology of the embryos. i.e. our results can not be explained by *Bmp7* always being expressed at lower levels than *Bmp4* (or vice versa). Additionally, as commented in above, we have now included an additional supplemental figure (Figure 2—figure supplement 3) that shows that how different patterns of *Bmp4* misexpression nonetheless drive pSmad1/5/8 to a specific level throughout the spinal cord. This series of experiments also shows that similar level of pSmad driven by different *Bmps* does not result in the same cellular identity, i.e. low [BMP4] → low pSmad results in a different series of cellular fates than high [BMP7] → low pSmad.

4) Some discussion of what has been done in the BMP field to match ligands with receptors-has anything been done? This concept that the different BMPs have specificity for receptors is reported. More reference to what has been done in the field along these lines seem relevant. Also, there is a whole literature on BMP gradients specifying cell type in zebrafish. Is there just a difference is species or has it not been addressed in experiments in fish. Work from Mary Mullins comes to mind.

We thank reviewer 2 for these suggestions, which we have addressed as follows:

a) We have added references to the classic biochemical studies that suggested that BMPs have differential binding affinities to type I BMP receptors and included references from the Mullins lab in the Discussion.

b) As commented previously, we think it an interesting and likely possibility that the “code” of BMPs used to specify cell type will differ – slightly? profoundly? – in different species. This question requires further study and is discussed further in the first paragraph of the Discussion.

We also discuss that we can only currently make a case about the role of BMPs in the context of spinal cord development – not about their activity in development of other species/organ systems. However, we note that was the spinal cord was one of the systems that supported the canonical views that the BMPs act as morphogens – more by analogy with Shh than by observation – making it critical to reassess how strong the evidence is that BMPs act as morphogens in other systems.

Reviewer #3:

The main crux of this study is that different BMPs act differentially to direct dorsal progenitor and interneuron identity and do not act as a set of classical morphogens, working through concentration and time to progenitor lineage restriction and specify cell fate.

The data seem reasonable for a differential effect on interneuron identity but I wasn't so convinced by the effects on progenitor identity.

The effects on progenitor identity are dramatic – both *Bmp4* and *Bmp7* expression increase the size of the Atoh1 domain, while *Bmp4* decreases the size of the Ngn1 domain. However, we did not make it clear enough in the previous version of the manuscript that it is critical to only compare the control vs. experimental electroporated sides of the spinal cord. We have now added a note to this effect to the figure legend of Figure 5

I also wonder whether a slightly different statistical analysis might not tease things out better and make the discussion easier- namely something like the Kolmagorov-Smirnov test of differences in distribution. My understanding is that different BMPs should have differing distributions of effects on the different dorsal interneuron populations and so an analysis of these distribution- rather than a direct comparison of one cell type by different BMPs might be a better way to go forward.

We thank reviewer 3 for this suggestion. As they note, the KS test is designed to examine whether there are differences in the manner in which experimental values are distributed, i.e. could two sets of numbers be taken from the same distribution (or not)? However, in general for our studies, it was more appropriate to determine the probability that the number of a particular class of cells, or level of gene expression, in our experimental conditions were similar to control.

In these tests, we assume equal variance because the experiments are performed under very similar conditions. Supporting this assumption, it made very little difference to the p values when we relaxed the equal variance assumption using the Welch-Satterthwaite approximation. We also tested for a normal distribution according to the Anderson-Darling test. Where the data fit a normal distribution, we used the t-test; where it did not, we used a Mann-Whitney test.

However, we were interested to assess whether we could determine whether *Bmp4* and *Bmp7* misexpression result in statistically similar (or not) range of cellular activities. We used the Fisher exact test on a 6x2 table, rather than the KS test, because the categories that we are assessing “# of RP cells, dI1s, dI2s, dI3s, dI4s, dI5” are indivisible biological entities not forming a ranked scale and are thus far from the continuous parameter required by the KS test. We found that the probability that cellular activities that result from *Bmp4* and *Bmp7* misexpression are from the same distribution is p< 0.0002. This analysis has been added to the figure legend of Figure 2 and further details added to the Materials and methods.

I go through each figure in turn with comments.

Figure 1. This seems completely reasonable in terms of an analysis of gene expression at different stages in the chick.

Thank you.

Figure 2 was slightly puzzled by the upregulation of phospho smad on the contralateral hemi-spinal cord with BMP4 electroporation in E.

We have considerable evidence that the effect of *Bmp* misexpression is non-automonous and is not confined to the electroporation side of the spinal cord. Thus, we think that the upregulation of pSmad on non-electroporated side is a result of diffusion of the ectopic BMP4 ligand.

I wasn't sure of how the authors distinguished dI4 from dI6 in their summary between N and O- indeed there is no mention of dI6 in the quantitation shown in V. Could it be possible that there is a dramatic reduction in dI6 with BMP4 or a dramatic increase in dI4 with BMP7?

This possibility is unlikely. While the images in Figure 2 might suggest that we used the spatial separation of the Pax2^+^ dI4 population from Pax2^+^ dI6 population to distinguish these cell types, in fact we also co-stained with antibodies against Bhlhb5, which labels a number of spinal cell types including the dI6s, but not the dI4s. Thus, the Pax2^+^ Lhx1/5^-^ Bhlhb5^-^ dI4s could be unambiguously distinguished from the Pax2^-^ Lhx1/5^+^ Bhlhb5^-^ dI2s and the Pax2^+^ Lhx1/5^+^ Bhlhb5^+^ dI6s. We are thus confident that there is no ambiguity in our cell count for dI4s and have clarified this issue in the legend to Figure 2.

However, we were concerned about ambiguity in our dI6 count since Pax2^+^ Lhx1/5^+^ Bhlhb5^+^ does not uniquely distinguish the dI6s; this is why we did not include any dI6 counts. There is unfortunately no unambiguous marker in chicken that would distinguish the dI6s – antibodies against Wt1 for example only work in mouse.

The results for dI3 and 5 look convincing. I did notice that the spatial organisation of these distinct interneurons also seemed perturbed- so not just number but position could also be altered by changing BMP expression.

In V- I wonder if an analysis of the distribution of interneuron identities following the different perturbations might be informative- i.e. is the distribution of RP and dorsal interneuron types changed differentially by BMP4 and BMP7 or are both distributions similar?

Reviewer 3 is correct. We have focused on the changes in cell numbers – but there are spatial organizational changes too, for the *Bmp4*, but not *Bmp7*, electroporations. In summary:

RP, dI1 and dI3 populations: for both the *Bmp4* and *Bmp7* electroporations, these populations expand but remain in the correct spatial order with respect to each other. Thus, the RP remains the most dorsal population, the dI1 is ventral to RP and the dI3s are ventral to the dI1s.

dI2 population: For the *Bmp4* electroporation only, the dI2s both expand and change location, such that they are intermingled with/ventral to the dI3 population. Interestingly, we see this shift in location occurring between the 25ng/μl and 50ng/μl condition in the *Bmp4* concentration series. Thus, for 5 ng/μl and 25 ng/μl conditions, the dI2 population is larger following *Bmp4* electroporation, but in the correct spatial order with respect to the dI1 and dI3s, i.e. midway between them. However, by the 50 ng/μl condition (which has >2-fold more dI2s than the 25 ng/μl, Figure 4) the dI2 are now ventral to dI3s. One hypothesis is that as the dP2 Ngn1^+^ domain is depleted, the more ventral Ngn1^+^ domain, that normally gives rise ventral INs, is co-opted to make dI2s.

We show this location change in the schematics depicted in Figure 2 and now make explicit mention of this result in the legend for the schematics, as well as discussing it in subsection “BMPs have distinct roles directing dorsal spinal fates”.

Figure 3 shows quantitation of RT q-PCR results on embryoid bodies treated with different BMPs suggesting that BMP4 upregulates dI1 markers preferentially, in comparison to the other BMP family members. This data looks convincing and the distributions (see above) look clearly different. I do wonder whether RT-1-PCR is the best measure of this though- rather than the protein expression (cellular) analysis shown in other parts of the paper.

We chose to use RT-PCR as a standardised readout of the effect of BMP recombinant protein on gene expression. This approach permitted us both to a) perform a relatively quickly analysis of the many conditions in our mESC experiments and b) overcome technical issues that we had with immunolabelling mouse EBs, which generally do not stain as well as either human EBs or embryonic tissue.

However, we did collect some immunohistochemical data in the early stages of this project and found that the RT-PCR quantification closely mirrored the cell counts. This data is not included in the study, but if the reviewer would like – we can mention this finding in the Materials and methods.

Figure 4. Here- changes to concentration of plasmids electroporated are used as a proxy for changes in BMP concentration. This assays, quite rightly, by phospho SMAD expression. It would be nice to see some kind of quantitation of the effects of the different plasmid concentrations on SMAD expression.

We thank the reviewer for this suggestion and have now included a quantification of a wider range of plasmid concentrations on R-Smad activity in Figure 4.

Both Smad1 and Smad5 are expressed broadly and at high levels in the embryonic spinal cord (see Hazen et al., 2012) making it unlikely that they are directly regulated by the BMPs. This possibility is supported by our mESC cultures: in our unpublished observations, we do find that the expression of Smad1 and Smad5 increases over time as mESCs differentiate into EBs. However, there is no significant difference in the Smad1/5 levels in the RA (control) vs. RA/BMP-treated (experimental) EBs.

Again, the distributions of dorsal interneuron identities generated by these different perturbations do look different but it might be nice to do this direct comparison through the same concentration of plasmid across the different interneuron types to see if the different concentrations display the same distributions.

Was this experiment performed in Figure 2 (and Figure 2—figure supplement 1)? In that study the same concentration of expression plasmid is electroporated for BMP4, BMP5, GDF7 and BMP7. We find that the same concentration of each of BMPs results in a range of different cellular activities. When we electroporated a range of concentrations of one of these BMPs, BMP4 in Figure 4, we find that we get the same range of cellular activities, but that BMP4 is more effective at higher concentrations.

Figure 5. Here- I wasn't really sure on the differences in progenitor characteristic and identity that were being displayed. The progenitor domains seemed the same, in terms of their distributions following BMP4 and BMP7 misexpression but I might be missing something subtle here.

We apologize to the reviewer. We did not make it clear that it is important to compare only the electroporated sides in the control vs. experimental spinal cords in Figure 5 because of the wide-ranging effects of *Bmp* misexpression. We have added a further explanatory note to the figure legend, as well as noting that the effect on the Ascl1 domain is not as profound as the effect on the Atoh1/Ngn1 domain as also requested by reviewer 1.

Figure 6. I wonder if different pulse-lengths of BrdU might be informative to look at changes in cell cycle following different BMP exposures.

This experiment is an excellent idea, and we plan to use such approaches in our follow up studies. Here we have focussed on assessing how BMPs alter cell cycle dynamics in the mESC model system, to further characterise differences/commonality in our model systems between the BMPs and their effects on cell cycle.

Finally, in Figure 7 – I wasn't clear on the expression of the BMPR1b in the dorsal spinal cord early in development. Is it expressed in the right time and right place to mediate differential signalling dependent on the BMP being exposed in the wild-type.

We have now added further in situ hybridization panels – Figure 7 – which show that while BmprIb is not expressed at HH stage 18, it is broadly expressed by HH stage 21, when dorsal neurons start to differentiate (timeline in Figure 1).

[Editors' note: the author responses to the re-review follow.]

Reviewer 3 was not satisfied with the claims regarding cell cycle exit and would like the following changes:

The authors should resist from making firm conclusions about the length of the cell-cycle and the ability to promote exit from the cell-cycle, and should tone down the text regarding these points. They should also remove Figure 6 and Figure 7 as these models are misleading. The models in Figure 8 is a problem, but as long as the text tones down the cell-cycle effects and states that other experiments are needed when discussing the final over-arching model, it will be OK.

We have removed the mention of cell cycle exit in the results and altered Figure 6 and Figure 7 such that they are now summaries of the data, i.e. how a given BMP alters progression through the cell cycle, without implying a functional consequence. The data is complex and we believe these summaries will help guide the reader. We have further clarified that the final model is our interpretation of the data, and added the statement similar to “other experiments are needed” to the manuscript.